# Video2GUI: Synthesizing Large-Scale Interaction Trajectories for Generalized GUI Agent Pretraining

Weimin Xiong [1 2 †]  Shuhao Gu [2]  Bowen Ye [2]  Zihao Yue [2 3]  Lei Li [2 4]
Feifan Song [1]  Sujian Li [1 ‡]  Hao Tian [2 ‡]

## Abstract

Recent advances in multimodal large language models have driven growing interest in graphical user interface (GUI) agents, yet their generalization remains constrained by the scarcity of large-scale training data spanning diverse real-world applications. Existing datasets rely heavily on costly manual annotations and are typically confined to narrow domains. To address this challenge, we propose Video2GUI, a fully automated framework that extracts grounded GUI interaction trajectories directly from unlabeled Internet videos. Video2GUI employs a coarse-to-fine filtering strategy to identify high-quality GUI tutorial videos and convert them into structured agent trajectories. Applying this pipeline to 500 million video metadata entries, we construct WildGUI, a large-scale dataset containing 12 million interaction trajectories spanning over 1,500 applications and websites. Pre-training Qwen2.5-VL and MiMo-VL on WildGUI yields consistent improvements of 5–20% across multiple GUI grounding and action benchmarks, matching or surpassing state-of-the-art performance. Project page: `https://weiminxiong.github.io/Video2GUI/`.

## 1. Introduction

Recent advances in multimodal large language models have underscored the importance of agents capable of autonomously interacting with graphical user interfaces (Zhang et al., 2024a; Wang et al., 2025; Qin et al.,

2025). Such agents can automate tasks on digital devices by perceiving visual interface states and emulating human actions, including clicking, typing, and dragging, across diverse platforms such as web, desktop, and mobile applications (Cheng et al., 2024; You et al., 2024). A key prerequisite for developing generalized GUI agents is access to large-scale and diverse trajectory data that can precisely document GUI interactions. These data capture authentic user behavior patterns across a wide range of applications, tasks and systems, and are essential for improving the generalization capability of agents (Li et al., 2024; Wu et al., 2024; Chen et al., 2025).

To acquire training data, most existing approaches rely heavily on manually annotated interaction trajectories (Deka et al., 2017; Rawles et al., 2023) or on simulated environments (Shvo et al., 2021; Lee et al., 2024). While these sources provide high-quality supervision, they entail substantial annotation costs and suffer from limited scalability, which in turn constrains the agent's ability to generalize to unseen interfaces and tasks (Gao et al., 2024). In contrast, internet videos constitute a rich repository of real-world GUI usage, providing detailed step-by-step demonstrations of software operations (Zhang et al., 2026). However, effectively leveraging such data presents significant challenges. The immense diversity of online videos makes it difficult to reliably filter high-quality instructional GUI recordings featuring superior visual fidelity, topical relevance, and instruction clarity. Moreover, as raw videos lack explicit interaction annotations, extracting structured action trajectories and precisely grounding them to corresponding screen coordinates remains a significant challenge.

To tackle the problems above, we propose **Video2GUI**, a scalable framework that can both filter high-quality GUI instructional videos from billions of internet videos and automatically annotate accurate GUI interaction trajectories (Figure 1). To effectively handle video data at internet scale, we adopt a coarse-to-fine filtering strategy. We begin with metadata-based textual filtering as a coarse screening step. This low-cost stage leverages video titles, descriptions, and keywords to quickly eliminate videos unrelated to GUI-based software operations, thereby avoiding un-

---

[†]Contribution during internship at Xiaomi LLM-Core Team.
[‡]Co-corresponding authors. [1]National Key Laboratory for Multimedia Information Processing, School of Computer Science, Peking University [2]LLM-Core, Xiaomi [3]Renmin University of China [4]The University of Hong Kong. Correspondence to: Sujian Li <lisujian@pku.edu.cn>.

*Proceedings of the 43rd International Conference on Machine Learning*, Seoul, South Korea. PMLR 306, 2026. Copyright 2026 by the author(s).

*Table 1.* Comparison with existing datasets. WildGUI is the largest open-source GUI pre-training dataset, offering comprehensive coverage across website, mobile, and desktop platforms with over 12M trajectories and 124M images.

| Dataset | Platform | | | Scale & Statistics | | | | Inst. Level |
|---|---|---|---|---|---|---|---|---|
| | Website | Mobile | Desktop | Environments | Instructions | Images | Turns | |
| MiniWoB++ (Liu et al., 2018) | ✓ | ✓ | ✗ | 114 | 100 | 17,971 | 3.6 | Low-level |
| MIND2WEB (Deng et al., 2023) | ✓ | ✗ | ✗ | 137 | 2,350 | 2,350 | 7.3 | High-level |
| AITW (Rawles et al., 2023) | ✗ | ✓ | ✗ | 357 | 30,378 | 715,142 | 6.5 | High & low |
| AndroidControl (Li et al., 2024) | ✗ | ✓ | ✗ | 833 | 14,538 | 15,283 | 4.8 | High & low |
| GUI-World (Chen et al., 2024) | ✓ | ✓ | ✓ | - | 12,379 | 83,176 | 6.7 | High-level |
| GUI-Odessey (Lu et al., 2025c) | ✗ | ✓ | ✗ | 201 | 7,735 | 118,791 | 15.4 | High-level |
| GUI-Act (Chen et al., 2025) | ✓ | ✗ | ✗ | 50 | 67k | 13k | 7.9 | Low-level |
| GUI-Net (Zhang et al., 2026) | ✓ | ✓ | ✓ | 280 | 1M | 1M | 4.7 | High-level |
| MONDAY (Jang et al., 2025) | ✗ | ✓ | ✗ | - | 20K | 313k | 15.7 | High-level |
| GUI-360° (Mu et al., 2025) | ✗ | ✗ | ✓ | 3 | 13,750 | 105,368 | 7.6 | High-level |
| **WildGUI (Ours)** | ✓ | ✓ | ✓ | 1,500+ | 12.7M | 124.5M | 9.7 | High & low |

necessary storage and computational overhead. However, filtering based solely on metadata cannot guarantee video or instructional quality. We therefore introduce a visual scoring model to perform fine-grained evaluation of video content, assessing factors such as visual fidelity, instructional completeness, and task relevance. This strategy effectively balances processing efficiency with the quality of the selected videos. To extract explicit interaction annotations from raw videos accurately, we first identify task instructions, operation timestamps, action details, and the underlying reasoning behind each action. We then extract screenshots at the identified timestamps and perform precise spatial grounding to map actions to exact screen coordinates, thereby mitigating the grounding difficulties caused by frame compression in videos (Mu et al., 2024).

Using this pipeline, we introduce **WildGUI**, a large-scale and diverse GUI pre-training dataset constructed from unlabeled web videos. Starting from 500 million video metadata entries, we extract 12.7 million GUI operation trajectories comprising 124.5 million screenshots, spanning more than 1,500 applications and websites (see Table 1). Compared to existing datasets, WildGUI offers a distinct advantage by sourcing diverse and comprehensive actionable training data for GUI agents directly from raw Internet-scale videos. Leveraging WildGUI, we can perform continual pre-training on Qwen2.5-VL and MiMo-VL to enhance their generalization ability across diverse applications and environments. Further, we conduct post-training on carefully curated open-source datasets to refine task-specific performance. We evaluate the resulting agents on GUI grounding tasks, as well as both offline and online agent benchmarks. Experimental results demonstrate that models pre-trained on WildGUI achieve consistent and substantial improvements across all evaluated benchmarks and model architectures, matching or surpassing state-of-the-art performance. These results confirm the effectiveness and scalability of our constructed dataset. We summarize our contributions as follows:

- We propose Video2GUI, a fully automated and scalable framework for extracting high-quality GUI interaction trajectories from large-scale unlabeled web videos.

- We construct WildGUI, a large-scale and diverse GUI pre-training dataset with 12 million trajectories spanning over 1,500 real-world applications and websites.

- Extensive experiments on MiMo-VL and Qwen2.5-VL show that pre-training on WildGUI significantly improves performance across multiple popular GUI grounding and agent benchmarks, demonstrating strong effectiveness and generalization.

- We will release the WildGUI dataset and the Video2GUI pipeline to facilitate future research on scalable GUI agent training and evaluation.

**Conflict of Interest Disclosure** The authors S.G., B.Y., Z.Y., L.L., and H.T. are employed by Xiaomi, which leads the development of MiMo-VL, one of the base models evaluated in this paper. W.X. conducted this work during an internship at Xiaomi.

## 2. Formulation of GUI Agent

We formulate the GUI agent interaction as a Partially Observable Markov Decision Process (POMDP), defined by the tuple $(\mathcal{U}, \mathcal{S}, \mathcal{A}, \mathcal{O}, \mathcal{T})$. Here, $\mathcal{U}$ denotes the space of high-level user instructions (e.g., task descriptions), $\mathcal{S}$ represents possible environment states, $\mathcal{A}$ is the action space consisting of atomic GUI operations (e.g., clicking, typing), and $\mathcal{O}$ denotes the observation space. The transition function $\mathcal{T} : \mathcal{S} \times \mathcal{A} \rightarrow \mathcal{S}$ models the evolution of GUI states in response to agent actions. In this context, $\mathcal{U}$ and $\mathcal{A}$ are typically expressed in natural language, while $\mathcal{O}$ consists of multimodal feedback such as screen screenshots. Under this framework, the task-solving process unfolds as a sequential

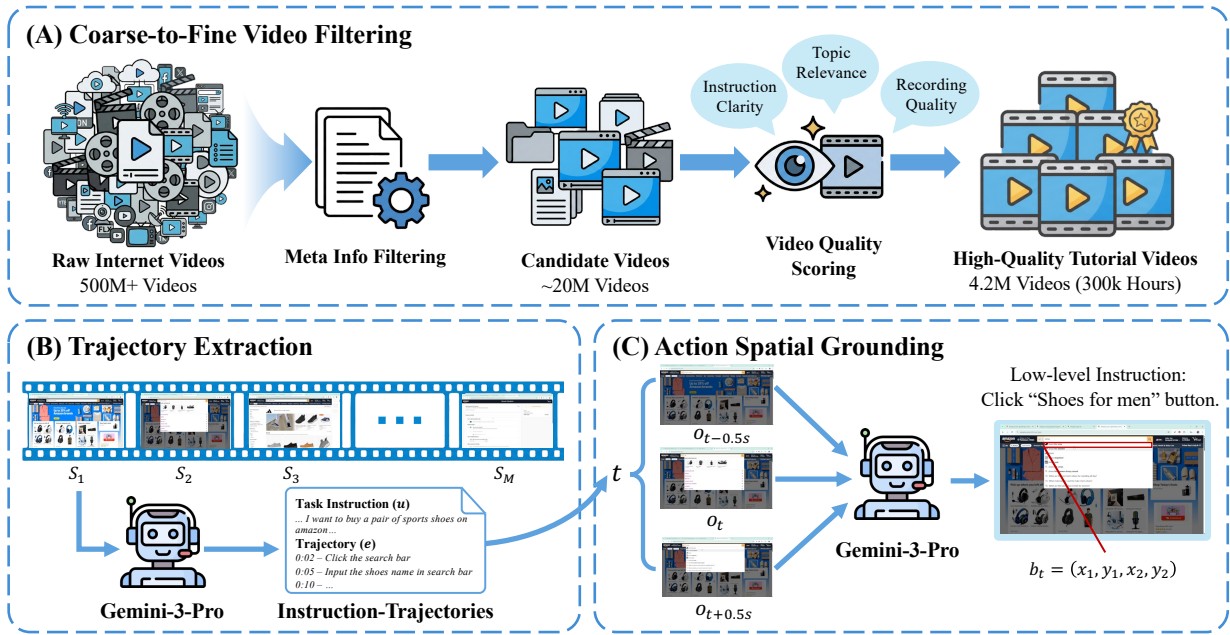

*Figure 1.* Overview of the Video2GUI pipeline. Video2GUI consists of three stages: (A) coarse-to-fine video filtering for selecting high-quality tutorial videos, (B) trajectory extraction that converts video segments into instruction–trajectories sequences, and (C) action spatial grounding that maps low-level instructions to precise UI targets to produce grounded trajectories.

decision-making loop where the agent interacts with the environment to fulfill the user instruction $u$.

At each time step $t$, the agent $\pi_\theta$ receives an observation $o_{t-1} \in \mathcal{O}$ and selects an action $a_t \in \mathcal{A}$ based on the policy $\pi_\theta(\cdot \mid e_{t-1})$, where $e_{t-1} = (u, a_1, o_1, \ldots, a_{t-1}, o_{t-1})$ denotes the interaction history. Each action $a_t$ is parameterized as a tuple $(\tau_t, b_t)$, where $\tau_t$ specifies the action type (e.g., click, type, scroll), and $b_t$ denotes the corresponding action parameters, such as click coordinates, target UI elements, or input text. Executing $a_t$ induces a state transition to $s_t \in \mathcal{S}$, yielding execution feedback $o_t \in \mathcal{O}$. This perception–action loop continues until the task is completed or a maximum step budget is reached, resulting in a trajectory $e_n = (u, a_1, o_1, \ldots, a_n, o_n)$ of length $n$.

## 3. Video2GUI

To curate GUI interaction data, we introduce the Video2GUI synthesis framework alongside a specialized training strategy. As illustrated in Figure 1, the Video2GUI pipeline converts raw internet videos into structured interaction trajectories $\mathcal{D} = \{(u, e)^{(i)}\}_{i=1}^{|\mathcal{D}|}$ through three progressive stages . We begin with coarse-to-fine video filtering (§ 3.1) to identify high-quality GUI tutorial content from internet videos. Next, we execute trajectory extraction (§ 3.2), deriving task instructions, action timestamps, action details, and action rationales from the videos. We then apply action space grounding (§ 3.3) to map the extracted actions to precise

screen coordinates. Finally, to fully leverage this large-scale synthetic data, we introduce a two-stage agent training strategy (§ 3.4) that combines continual pre-training on Video2GUI data with supervised post-training to equip the agent with robust execution capabilities.

### 3.1. Coarse-to-Fine Video Filtering

**Meta Info Filtering** Given the massive scale and heterogeneity of internet videos, directly processing raw video content is prohibitively expensive. For instance, downloading and storing hundreds of millions of videos would require hundreds of petabytes of storage, while blind processing would waste substantial computational resources on irrelevant content such as daily vlogs or news commentary. To ensure the extraction of high-quality GUI tutorial videos, we first conduct a rapid coarse filtering based on video metadata. We begin by collecting metadata from over 500 million YouTube videos obtained through a combination of public repositories and large-scale web crawling. We choose YouTube as the data source because it offers excellent diversity, covering software tutorials across multiple languages, countries, and platforms, which naturally provides cross-lingual and cross-cultural GUI interaction data. The collected metadata includes video titles, authors, upload timestamps, keywords, and textual descriptions. To balance filtering quality and computational cost, we leverage DeepSeek-V3 (Liu et al., 2024) to annotate relevance labels between video metadata and GUI operation content

for 10K samples, and subsequently fine-tune a lightweight Qwen2.5-7B (Yang et al., 2024) with a classification head to enable scalable inference. We validate the effectiveness of this automated annotation against a held-out set of manually labeled samples, where DeepSeek-V3 demonstrates high alignment with human judgment. We apply upsampling to this synthetic dataset to ensure balanced positive and negative examples, and fine-tune Qwen2.5-7B accordingly. The resulting model is then applied to classify all collected video metadata. Through this metadata-based coarse filtering round, we reduce the candidate videos to approximately 20 million. For detailed information on the classification model training process and prompts, please refer to the Appendix A.

**Video Quality Scoring** Although metadata-based coarse filtering effectively removes videos irrelevant to GUI interactions, it remains insufficient for ensuring high-quality instructional content. For example, some videos focus on software introductions or advertisements rather than concrete operational demonstrations, while others suffer from poor recording quality, including low resolution or missing narration. These issues cannot be reliably detected using metadata alone. To address this limitation, we introduce a fine-grained, content-based video scorer that directly analyzes video content.

Specifically, we leverage an omnimodal model that supports diverse inputs including text, video, and audio to assess video quality. For each candidate video, we extract the first minute and evaluate it along three complementary dimensions. Topic relevance measures whether the video focuses on teaching GUI operations on the target platform and favors content that explains GUI functionalities or demonstrates specific tool usage. Instruction clarity assesses the clarity and coherence of instructional narration and guidance. Screen recording quality evaluates whether the visual content is clear, complete, and stable enough to support effective learning of GUI operations. To reduce annotation costs, we curate approximately 200 hours of videos sampled from the coarsely filtered set and employ Gemini 3 Pro (Comanici et al., 2025) to annotate each video with dimension-wise scores and corresponding rationales. We then fine-tune a lightweight Qwen2.5-Omni (Xu et al., 2025) as a video-level quality scorer. Applying the trained model to 20 million coarsely filtered videos, we retain 4.16 million videos, yielding approximately 300,000 hours of high-quality GUI instructional content. Further details on the model training procedure and annotation prompts are provided in Appendix B.

## 3.2. Trajectory Extraction

In the previous stage, we obtained a large collection of high-quality GUI tutorial videos through coarse-to-fine filtering.

However, these raw videos do not directly provide the structured supervision signals required to train GUI foundation models. The goal of this stage is to transform unstructured videos into task-oriented instruction–trajectory pairs, thereby extracting learnable GUI interaction data. Formally, given an input video $V$, our objective is to extract a set of instruction-trajectory pairs as defined in Section 2:

$$\mathcal{D}(V) = \{(u^{(k)}, e^{(k)})\}_{k=1}^N \qquad (1)$$

where each pair $(u^{(k)}, e^{(k)})$ corresponds to the $k$-th independent task instance in the video. Here, $u^{(k)}$ denotes the natural language instruction for the task, while $e^{(k)}$ represents the corresponding interaction trajectory, in which each action step is associated with a precise timestamp.

To achieve this, we employ Gemini-3-Pro directly as the annotation model. Since long videos pose challenges to the model's context window, we adopt a sliding-window strategy augmented with historical context memory. Specifically, each input is constrained to a video segment of at most 4 minutes. Videos exceeding this duration are partitioned into a sequence of consecutive segments $\{S_1, S_2, \ldots, S_M\}$. When processing the $j$-th segment $S_j$, the model receives the current video frames alongside the extracted results from preceding segments $\mathcal{D}(S_{1:j-1})$, which serve as textual context. This design enables the model to maintain long-range memory across segments, allowing it to accurately recognize tasks that span segmentation boundaries or depend on prior operations. Compared to foreground–background detection in TongUI (Zhang et al., 2026) or inverse-dynamics-based methods in VideoAgentTrek (Lu et al., 2025a) that rely on low-level visual cues and short-term inference, our VLM-driven approach supports long-horizon reasoning and captures the underlying intent of actions. During trajectory extraction, we also require the model to output a visually grounded textual description, the low-level instruction, for each interaction step in the trajectory $e^{(k)}$. Please refer to Appendix C for more details about the annotation process.

## 3.3. Action Spatial Grounding

When extracting trajectories from videos, the input videos are compressed to accommodate long-context processing, which inherently limits the visual resolution required for precise pixel-level localization. Therefore, we introduce the action spatial grounding stage that maps the extracted actions to high-resolution screenshots. For each interaction action extracted at timestamp $t$, we retrieve a triplet of high-resolution frames $O_t = \{o_{t-0.5s}, o_t, o_{t+0.5s}\}$ from the raw video. This multi-frame input is crucial for handling rapid visual changes caused by the high-frequency nature of GUI interactions[1]. For each frame, we employ Gemini-3-Pro to

---

[1]We empirically set the temporal offset to 0.5 seconds, roughly matching the average duration of a single GUI action.

*Table 2.* Performance comparison on **ScreenSpot-Pro** (Li et al., 2025) and **OSWorld-G** (Xie et al., 2025). (↑ Gain) indicates the improvement over the base model. After continual pre-training on WildGUI, the base models match or surpass state-of-the-art performance on GUI grounding evaluation. Results marked with '*' are evaluated by us.

| Agent Model | ScreenSpot-Pro | | | OSWorld-G | | | | |
|---|---|---|---|---|---|---|---|---|
| | Text | Icon | Avg | Text Match. | Elem. Rec. | Layout Und. | Fine-grained | Avg |
| *Proprietary Models* | | | | | | | | |
| Gemini-2.5-Pro | - | - | 11.4 | 59.8 | 45.5 | 49.0 | 33.6 | 45.2 |
| Seed1.5-VL | - | - | 60.9 | 73.9 | 66.7 | 69.6 | 47.0 | 62.9 |
| *Open-Source Models* | | | | | | | | |
| Qwen3-VL-2B* (Bai et al., 2025a) | 56.1 | 18.9 | 41.9 | 61.7 | 45.8 | 54.2 | 39.6 | 45.9 |
| GTA1-7B (Yang et al., 2025) | 65.5 | 25.3 | 50.1 | 42.1 | 65.7 | 62.7 | 56.1 | 55.1 |
| UI-Venus-7B (Gu et al., 2025) | 67.1 | 24.3 | 50.8 | 74.6 | 60.5 | 61.5 | 45.5 | 58.8 |
| OpenCUA-7B (Wang et al., 2025) | - | - | 50.0 | - | - | - | - | 55.3 |
| GUI-Owl-7B (Ye et al., 2025) | 69.4 | 31.5 | 54.9 | 64.8 | 63.6 | 61.3 | 41.0 | 55.9 |
| Qwen3-VL-8B* (Bai et al., 2025a) | 67.6 | 21.3 | 49.9 | 69.0 | 55.5 | 59.7 | 47.7 | 54.8 |
| Qwen3-VL-32B* (Bai et al., 2025a) | 73.4 | 25.0 | 54.9 | 72.8 | 63.3 | 66.4 | 51.7 | 60.6 |
| UI-TARS-72B (Qin et al., 2025) | 50.9 | 17.5 | 38.1 | 69.4 | 60.6 | 62.9 | 45.6 | 57.1 |
| *Effectiveness of WildGUI (Ours)* | | | | | | | | |
| Qwen2.5-VL-7B (Bai et al., 2025b)* | - | - | 26.8 | 41.4 | 28.8 | 34.8 | 13.4 | 27.3 |
| + WildGUI | 57.0 | 17.6 | 41.9 (↑15.1) | 70.0 | 54.6 | 57.7 | 46.2 | 53.7 (↑26.4) |
| MiMo-VL-7B (Xiaomi, 2025) | 55.7 | 18.4 | 41.2 | 65.0 | 59.2 | 59.0 | 40.2 | 54.7 |
| + WildGUI | 70.1 | 33.6 | **56.9** (↑15.7) | 80.8 | 68.3 | 71.1 | 61.4 | **67.6** (↑12.9) |

determine, based on the low-level instruction, obtained in the trajectory extraction stage, whether the action parameters can be localized from that frame, and to predict the precise grounding target $b_t$ (e.g., a bounding box or screen coordinates). The resulting grounded action is defined as:

$$b_t = g_\phi(o_{t-0.5s}, o_t, o_{t+0.5s}, \tau_t). \quad (2)$$

Among the three candidate frames, we select the first frame for which the model successfully produces a valid spatial grounding result as the final output. Manual verification of 200 randomly sampled actions confirms that over 95% are accurately parameterized using this strategy. Please refer to Appendix D for more details about the annotation process, and Appendix G for the API cost of Video2GUI pipeline.

### 3.4. Agent Training

To evaluate the effectiveness of the proposed Video2GUI framework, we follow prior work (Xu et al., 2024; Wang et al., 2025) and adopt a two-stage training strategy.

**Stage 1: Continual Pre-training** In the first stage, we pretrain the model on our large-scale WildGUI dataset to enhance its GUI interaction capabilities and generalization across diverse applications and environments. We design three complementary pretraining tasks: GUI grounding, GUI action prediction, and GUI trajectory modeling, and optimize the model with the mixed objective.

Specifically, the model is trained to: (i) *localize target UI elements* by predicting the corresponding coordinates or bounding boxes; (ii) *predict GUI actions* from a single

screenshot $o_t$ conditioned on a task-level instruction $u$; and (iii) *model multi-turn interactions* by autoregressively predicting actions from chronologically arranged sequences of screenshots and interaction histories, where the loss is computed exclusively on text tokens. The overall pretraining objective aggregates these three tasks as follows:

$$\mathcal{L}_{\text{pretrain}} = \mathcal{L}_{\text{ground}} + \mathcal{L}_{\text{action}} + \mathcal{L}_{\text{traj}}. \quad (3)$$

We conduct continual pretraining for one epoch, covering approximately 200 billion tokens.

**Stage 2: Post-training** In the second stage, we fine-tune the pretrained model on curated high-quality open-source datasets. This stage aims to consolidate the agent's policy by leveraging cleaner and more precise human supervision signals, which improves performance on domain-specific downstream tasks. We fine-tune the model for three epochs, totaling approximately 15 billion tokens. The datasets used in the post-training stage, as well as the prompt formats adopted in both stages are described in detail in Appendix F.

## 4. Experiments

### 4.1. Experiment Setup

**Implementation Details.** We implement our models using the Megatron framework to facilitate efficient large-scale distributed training. We optimize the model parameters using the AdamW optimizer with a weight decay of 0.05 and utilize a cosine learning rate schedule. To ensure the model captures the fine-grained visual details essential for accurate GUI operation, we configure the maximum number of

*Table 3.* Performance comparison on **AndroidControl** (Li et al., 2024) and **CAGUI** (Zhang et al., 2025) benchmarks. (↑ **Gain**) indicates the improvement over the base model. After continual pre-training on WildGUI, the base models match or surpass state-of-the-art performance on offline GUI agent evaluation. Results marked with '*' are evaluated by us.

| Models | AndroidControl-Low | | AndroidControl-High | | CAGUI | |
|---|---|---|---|---|---|---|
| | Type Acc. | Step SR | Type Acc. | Step SR | Type Acc. | Step SR |
| *Closed-source Models* | | | | | | |
| GPT-4o (Hurst et al., 2024) | 74.3 | 19.4 | 66.3 | 20.8 | 3.7 | 3.7 |
| *Open-source Models* | | | | | | |
| OS-Genesis-7B (Sun et al., 2025) | 90.7 | 74.2 | 65.9 | 44.4 | 38.1 | 14.5 |
| OS-Atlas-7B (Wu et al., 2024) | 73.0 | 67.3 | 70.4 | 56.5 | 81.5 | 55.9 |
| Aguvis-7B (Xu et al., 2024) | 93.9 | 89.4 | 65.6 | 54.2 | 67.4 | 38.2 |
| UI-TARS-7B (Qin et al., 2025) | 98.0 | 90.8 | 83.7 | 72.5 | 88.6 | 70.3 |
| *Effectiveness of WildGUI (Ours)* | | | | | | |
| Qwen2.5-VL-7B* (Bai et al., 2025b) | 94.1 | 85.0 | 75.1 | 62.9 | 74.2 | 55.2 |
| + WildGUI | 94.9 (↑0.8) | 90.3 (↑5.3) | 74.6 | 64.5 (↑1.6) | 88.3 (↑14.1) | 65.4 (↑10.2) |
| MiMo-VL-7B (Xiaomi, 2025) | 92.9 | 87.9 | 76.3 | 65.6 | 82.2 | 63.4 |
| + WildGUI | **95.5** (↑2.6) | **91.8** (↑3.9) | **80.6** (↑4.3) | **71.4** (↑5.8) | **90.3** (↑8.1) | **71.0** (↑7.6) |

input visual tokens to 4,096. Furthermore, to accommodate the extensive context required for long-horizon interaction trajectories, we set the maximum sequence length to 32,768 across both training stages.

**Training Configurations.** The training procedure consists of two stages. In Stage 1, the model is trained for 24,000 steps. The language model is optimized with a learning rate initialized at $2.5 \times 10^{-5}$ and linearly decayed to $1.0 \times 10^{-5}$, while the vision encoder is trained with a fixed learning rate of $8.0 \times 10^{-6}$. In Stage 2, the model is further refined on a curated dataset for 3 epochs, corresponding to 2,000 training steps. To facilitate stable fine-tuning, we adopt smaller learning rates in this stage: the language model learning rate decays from $1.0 \times 10^{-5}$ to $1.0 \times 10^{-6}$, and the vision encoder learning rate decays from $8.0 \times 10^{-6}$ to $8.0 \times 10^{-7}$. All experiments are conducted on a high-performance computing cluster equipped with 160 CPU cores, 512 GB of system memory, and 256 NVIDIA GPUs.

### 4.2. Evaluation

We evaluate our models across GUI grounding, offline agent evaluation, and online agent evaluation benchmarks. For GUI grounding, we utilize OSWorld-G (Xie et al., 2025) and ScreenSpot-Pro (Li et al., 2025). For offline agent evaluation, we employ AndroidControl (Li et al., 2024) to assess planning and execution capabilities in mobile environments, and CAGUI (Zhang et al., 2025) to evaluate performance within Chinese-language interfaces. For online agent evaluation, we use OSWorld (Xie et al., 2024) for desktop applications and AndroidWorld (Rawles et al., 2024) for mobile operating systems. Details are in Appendix H.

### 4.3. Main Results

**GUI Grounding Evaluation** We compared our WildGUI-pretrained models against both proprietary models and leading open-source models, demonstrating strong grounding performance despite their compact model size. As shown in Table 2, large-scale pre-training on WildGUI yields strong GUI grounding capabilities. Specifically, MiMo-VL-7B trained on WildGUI achieves state-of-the-art performance on OSWorld-G, attaining an average score of 67.6, surpassing both Qwen3-VL-32B at 60.6 and Seed1.5-VL at 62.9. The model exhibits consistent improvements across all sub-metrics, including element recognition and layout understanding. On the high-resolution ScreenSpot-Pro benchmark, Qwen2.5-VL-7B pre-trained on WildGUI also substantially improves upon the base model, advancing from 26.8 to 41.2, while MiMo-VL-7B achieves 56.9, outperforming the previous best open-source model Qwen3-VL-32B at 54.9, and ranking second only to Seed1.5-VL at 60.9. These results demonstrate that pre-training on WildGUI can effectively enhance the grounding capabilities of GUI agents across different model architectures.

**Offline GUI Agent Evaluation** We further assessed the planning and execution capabilities of our agents using the AndroidControl and CAGUI benchmarks. As presented in Table 3, our models achieve consistent performance gains across both high-level planning and low-level execution tasks. Notably, MiMo-VL-7B pre-trained on WildGUI attains a step success rate of 71.4 on AndroidControl-High and 91.8 on AndroidControl-Low, significantly outperforming the base model scores of 65.6 and 87.9, respectively. The benefits of scaling with diverse training data are also ev-

ident in the cross-lingual CAGUI benchmark, where MiMo-VL-7B achieves a type accuracy of 90.3 and a step success rate of 71.0, surpassing the base model's performance of 74.2 and 55.2. Similar improvements are observed for Qwen2.5-VL. These results demonstrate that WildGUI enhances atomic action execution and enables generalization across different languages and diverse application scenarios.

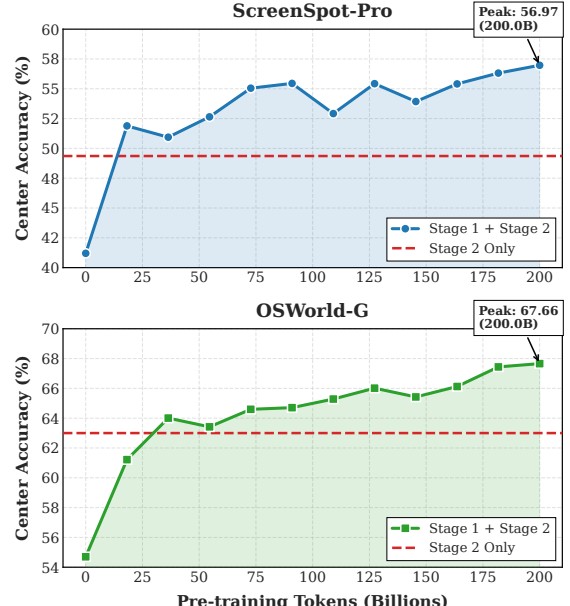

*Figure 3.* Impact of scaling pre-training data on performance.

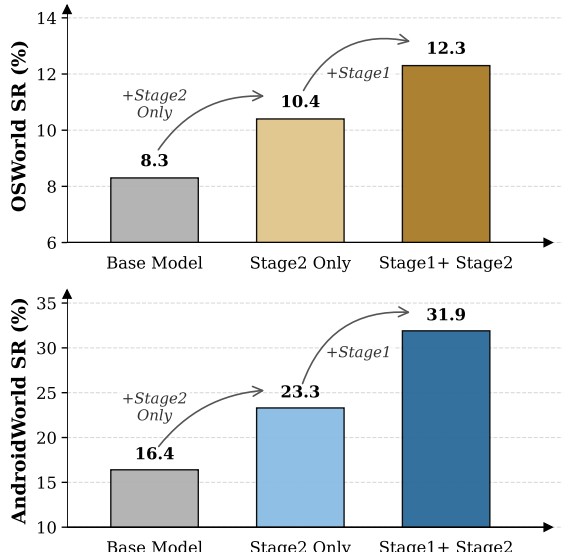

*Figure 2.* Performance on OSWorld and AndroidWorld. WildGUI demonstrates significant improvements over baseline models.

**Online GUI Agent Evaluation** To validate real-world applicability in dynamic environments, we evaluate MiMo-VL-7B on OSWorld and AndroidWorld. As shown in Figure 2, incorporating Stage1 pre-training on WildGUI yields substantial performance improvements compared to relying solely on Stage2 post-training with open-source data. Notably, despite WildGUI consisting entirely of offline GUI interaction data, the observed gains on both benchmarks demonstrate strong generalization to online, open-ended environments. On AndroidWorld, the full Stage1 + Stage2 pipeline achieves a success rate of 31.9%, nearly doubling the base model's 16.4% and significantly outperforming the Stage2-only baseline of 23.3%. A similar trend emerges on OSWorld, where our model reaches 12.3%, compared to 10.4% for the Stage2-only training. These results reveal that while supervised post-training remains important, large-scale pre-training on diverse offline GUI data provides a critical foundation for generalizing to real-world, dynamic GUI tasks and establishes a robust starting point for subsequent reinforcement learning in online environments.

## 5. Analysis

### 5.1. Scaling Effects

To investigate the impact of pretraining data scale on model performance, we conducted a scaling study by varying the number of pretraining tokens from 0 to 200 billion. We evaluated the models on two benchmarks, ScreenSpot-Pro and OSWorld-G, comparing against the "Stage 2 Only" baseline post-trained exclusively on open-source data. As illustrated in Figure 3, we observe a strong positive correlation between the scale of pretraining data and downstream task accuracy. On ScreenSpot-Pro, performance consistently improves with increasing pretraining tokens. The model starts at approximately 41% accuracy and reaches a peak of 56.9% at 200 billion tokens, significantly outperforming the Stage-2 Only baseline. Similarly, on OSWorld-G, accuracy improves from approximately 55% to a peak of 67.6%, with notable gains as data scale approaches 200 billion. Remarkably, the model surpasses the Stage-2 Only baseline at around 50 billion tokens and continues to show an upward trend without evident saturation. These results demonstrate a strong positive correlation between data quantity and agent performance, highlighting the importance of large-scale, diverse GUI data for model generalization.

### 5.2. Ablation Studies

To demonstrate the contribution of our pretraining objectives and the necessity of the two-stage paradigm, we conducted an ablation study on MiMo-VL-7B. As shown in Table 4, w/o $\mathcal{L}_x$ denotes the removal of specific loss components

during pretraining, while w/o Stage 2 represents the model trained without post-training. Our results reveal that different objectives impact distinct agent capabilities. Models trained without $\mathcal{L}_{traj}$ maintain competitive performance on static tasks (ScreenSpot-Pro and CAGUI) but exhibit significant drops on AndroidWorld (31.9 → 24.1). This indicates that modeling GUI interaction trajectories is crucial for long-horizon planning, as the trajectory loss enables the model to track goal states and maintain awareness of historical interactions during multi-step execution. Removing grounding supervision (w/o $\mathcal{L}_{ground}$) causes substantial degradation on ScreenSpot-Pro (56.9 → 49.8), confirming that $\mathcal{L}_{ground}$ is essential for accurate action grounding. This validates our design decision to incorporate explicit grounding supervision during pretraining. The w/o Stage 2 setting shows catastrophic performance drops across all metrics, especially on AndroidWorld (6.0). While first-stage pretraining on WildGUI provides broad GUI knowledge, the model requires alignment through high-quality second-stage data to perform well in complex instruction-following scenarios.

## 5.3. Data Quality Check

To validate the effectiveness of our filtering strategy and assess overall dataset quality, we conducted a user study in which five expert participants rated 300 sampled data points on a scale from 1 to 5. The participants are CS master's or Ph.D. candidates with prior research experience in VLM-based GUI agents and not involved in this project; each must pass a 20-sample qualification trial with ≥0.85 accuracy. As shown in Figure 4, for video quality evaluation, participants scored videos from different sources based on factors such as task relevance and screen recording quality. Our pipeline progressively improved the average score from 1.22 to 2.12 and finally to 4.45 through meta info filtering and video scoring respectively. For trajectory quality evaluation, participants compared our extracted trajectories with those from TongUI and VideoAgentTrek along three criteria: (1) *Accuracy* – whether actions are correctly identified with proper timestamps and spatial coordinates; (2) *Diversity* – the richness of platforms and task types within randomly sampled data; and (3) *Relevance* – whether trajectories reflect meaningful real-world GUI tasks. The five evaluators achieved a Krippendorff's $\alpha$ of 0.84, indicating strong inter-rater agreement. WildGUI achieved the highest overall score of 4.62, outperforming the two baselines at 3.35 and 4.05 respectively, thereby confirming the quality and diversity of WildGUI.

## 6. Related Work

### 6.1. GUI Agents

Recent advances in artificial intelligence have driven interest in developing agents capable of interacting with graphical

*Table 4.* Ablation study on training tasks for MiMo-VL-7B.

| Setting | ScreenSpot-Pro | CAGUI | AndroidWorld |
|---|---|---|---|
| **Ours** | **56.9** | **71.0** | **31.9** |
| w/o $\mathcal{L}_{ground}$ | 49.8 | 69.8 | 28.4 |
| w/o $\mathcal{L}_{action}$ | 50.5 | 65.3 | 27.6 |
| w/o $\mathcal{L}_{traj}$ | 54.6 | 70.2 | 24.1 |
| w/o Stage 1 | 49.3 | 64.2 | 23.3 |
| w/o Stage 2 | 28.2 | 45.7 | 6.0 |

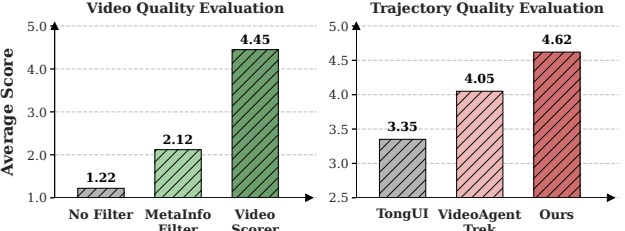

*Figure 4.* Average scores from human evaluation.

user interfaces (GUI) to perform complex tasks automatically (Qin et al., 2025; Zhou et al., 2025). Early GUI agents rely on structured data such as HTML or accessibility trees (Deng et al., 2023; Gur et al., 2023; Yang et al., 2023; He et al., 2024; Lai et al., 2024). With the advancement of MLLMs, vision-based methods have emerged (Hong et al., 2024; Zhang & Zhang, 2024; Zhang et al., 2024b), enabling direct GUI interaction via screenshots and natural language instructions. Building on these foundations, recent works (Qin et al., 2025; Yan et al., 2025; Zeng et al., 2025) have achieved state-of-the-art performance through innovations in training strategies and agent design. UI-TARS (Qin et al., 2025) develops a native end-to-end GUI agent via multi-stage post-training, enhancing perception, action and reasoning capabilities. ARPO (Lu et al., 2025b) adopts an end-to-end reinforcement learning paradigm, leveraging a replay buffer to reuse successful interaction experiences across training iterations. Despite these advances, such methods remain heavily dependent on large-scale and diverse training data, which is prohibitively expensive and difficult to collect manually from scratch.

### 6.2. Data Collection for GUI Agents

A key obstacle in building effective GUI agents lies in obtaining large-scale, diverse, and high-quality training data. Commonly used data collection schemes include human annotation (Deng et al., 2023; Li et al., 2024) and model-based synthesis (Li et al., 2024; Lu et al., 2025c), which typically provide supervision over planning processes, action sequences, and interaction targets of GUI agents (Cheng et al., 2024; Lu et al., 2024). However, existing datasets remain limited in scale and diversity, thereby constraining the progress of GUI agents. Given the abundance of GUI

operation resources available on the Internet, recent studies have explored harvesting GUI interaction trajectories directly from the Web to reduce annotation costs while maintaining data quality (Lu et al., 2025a; Zhang et al., 2026). However, existing approaches are largely confined to a single platform, such as mobile (Jang et al., 2025) or desktop environments (Song et al., 2025), and struggle to scale due to their reliance on keyword-based retrieval, which fundamentally constrains coverage and diversity. In contrast, Video2GUI adopts a scalable, top-down pipeline to directly filter and annotate raw YouTube videos, enabling a large-scale and diverse GUI dataset beyond prior methods.

## 7. Conclusion

In this paper, we address the challenge of data scarcity in GUI agent training by introducing Video2GUI, a fully automated framework that synthesizes high-quality interaction trajectories from unlabeled internet videos. Through coarse-to-fine filtering and VLM-driven trajectory extraction, we construct WildGUI, the largest GUI pre-training dataset to date with 12 million trajectories across 1,500+ applications and websites. Our experiments demonstrate that pre-training on WildGUI enhances model generalization, with Qwen2.5-VL and MiMo-VL achieving consistent performance improvements across GUI grounding and agentic tasks. These results validate that scaling training with diverse, offline video data provides a promising pathway toward generalized GUI agents. We hope the release of the WildGUI dataset and the Video2GUI pipeline will facilitate future research on more capable autonomous agents.

## Acknowledgement

We thank anonymous reviewers for their helpful comments on this paper. This work was partially supported by National Natural Science Foundation of China project (No. 62476010).

## Impact Statement

This paper presents work whose goal is to advance the field of Machine Learning. There are many potential societal consequences of our work, none of which we feel must be specifically highlighted here.

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

## A. Meta Info Classification Details

To determine whether a video is relevant to GUI operation tutorials based on its metadata, we construct a classification model that operates solely on textual information. Specifically, the model takes as input the video title, description, keywords, channel name, video category, and available subtitles when present. To obtain high-quality supervision for training, we leverage DeepSeek (Liu et al., 2024) to annotate the training data. For each video, we provide the complete set of meta information and prompt the model to output a binary label indicating whether the video is related to GUI operation tutorials (`true` or `false`), along with a brief reasoning explaining its decision. The outputs are constrained to a JSON format to facilitate automatic parsing and downstream processing. The detailed prompt used for data annotation is shown in Prompt 1. The inclusion of reasoning content allows us to iteratively refine and optimize the prompt design.

When training Qwen2.5-7B as the meta info classifier, we prioritize inference efficiency. Rather than prompting the model to generate complete classification results and rationales, we attach a lightweight binary classification head to the final hidden layer of Qwen2.5-7B. The model is guided by a minimal prompt—*"Classify based on the following content"*—and directly outputs class probabilities. The classifier is optimized using cross-entropy loss.

$$\mathcal{L}_{\text{CE}} = -\frac{1}{N} \sum_{i=1}^{N} [y_i \log(\hat{y}_i) + (1 - y_i) \log(1 - \hat{y}_i)] \tag{4}$$

where $N$ is the number of training samples, $y_i \in \{0, 1\}$ is the ground-truth label for sample $i$, and $\hat{y}_i \in [0, 1]$ is the predicted probability that the video is a GUI tutorial. We randomly sample 10K videos and annotate them using DeepSeek, yielding approximately 400 positive samples and over 9K negative samples. To mitigate the severe class imbalance, we apply upsampling to the positive samples to construct a balanced training set. The Qwen2.5-7B classifier is trained for 3 epochs with a learning rate of $2 \times 10^{-5}$.

## B. Video Scoring Details

To further refine the selection of high-quality GUI tutorial videos based on video content, we construct a video scoring model that takes the video as input and outputs the quality score. We evaluate videos across multiple dimensions. The detailed scoring criteria for each dimension are presented in Table 5. To obtain high-quality supervision for training, we leverage Gemini-3-Pro to annotate the training data. For each video, we provide the first minute of content (or the entire video if shorter than one minute) and prompt the model to output scores according to the criteria in Table 5, along with brief reasoning explaining its decision. The outputs are constrained to a JSON format to facilitate automatic parsing and downstream processing. The detailed prompt used for data annotation is shown in Prompt 2.

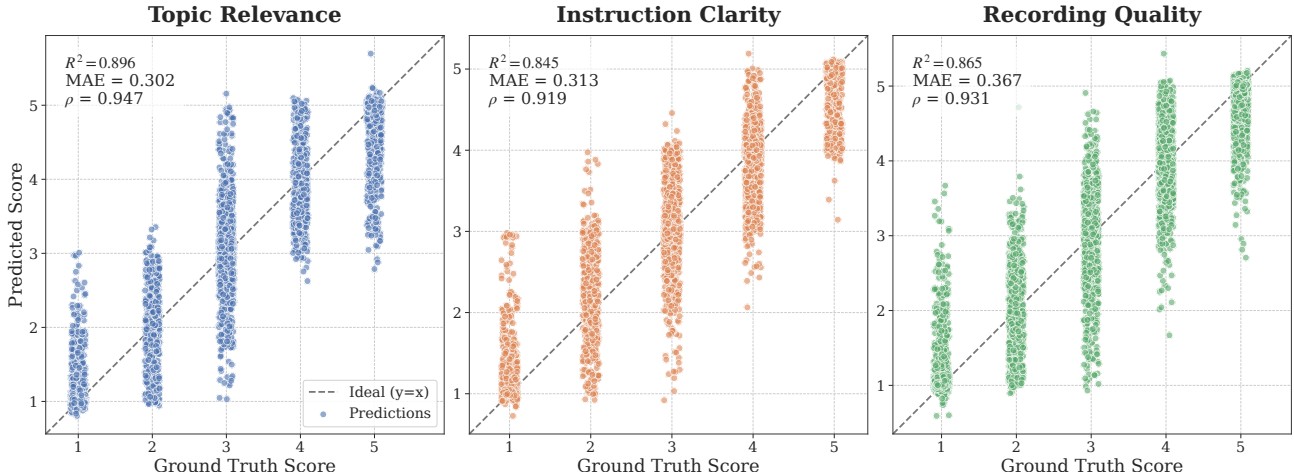

*Figure 5.* Performance of the video scoring model on the test set across three quality dimensions.

When training Qwen2.5-Omni as the video scoring model, we similarly prioritize inference efficiency. We attach three regression heads to the final hidden layer of Qwen2.5-Omni, corresponding to the three scoring dimensions. The model is

guided by a minimal prompt—*"Classify based on the following video"*—and is optimized using mean squared error (MSE) loss. The MSE loss is defined as:

$$\mathcal{L}_{\text{MSE}} = \frac{1}{N} \sum_{i=1}^{N} \sum_{j=1}^{3} (y_{ij} - \hat{y}_{ij})^2 \tag{5}$$

where $N$ is the number of training samples, $y_{ij}$ is the ground-truth score for sample $i$ on dimension $j$, and $\hat{y}_{ij}$ is the predicted score. The Qwen2.5-Omni scorer is trained for 3 epochs with a learning rate of $2 \times 10^{-5}$. We annotate 200 hours of training data using Gemini-3-Pro and apply upsampling to ensure balanced score distribution as much as possible. We set aside 100 videos as our test set, with evaluation results shown in Figure 5. As shown in the figure, our scorer demonstrates strong alignment with ground truth annotations across all three dimensions. Through manual verification, we establish quality thresholds requiring videos to achieve scores of at least 4.2 on all three dimensions—topic relevance, instruction clarity, and recording quality. Applying the trained model to 20 million coarsely filtered videos, we retain 4.16 million videos, yielding approximately 300,000 hours of high-quality GUI instructional content.

## C. Trajectory Extraction Details

Following prior work (Xie et al., 2024; Zhang et al., 2025), we adopt two distinct action spaces for mobile and desktop platforms, as detailed in Table 6. For the filtered high-quality videos, we selected only those with durations under 12 minutes based on metadata information, resulting in a total of 300,000 hours of annotated content.

We segment videos into 4-minute clips for annotation. For each segment, we prompt the model to annotate the following elements: (1) **User Task Instruction**: the user's intended task; (2) **Dense Caption**: a detailed description of the task completion process; (3) **Task Plan**: the step-by-step planning for task execution; (4) **Platform**: the operating system environment, including Windows, iOS, Android, etc.; (5) **Application Type**: the software used to complete the task (if applicable), such as WeChat, Google Chrome, etc.; (6) **Website Name**: the specific website visited (if the task is performed within a browser); and (7) **Action Trajectory**: the sequence of user operations performed during task completion.

For action trajectories, we annotate each action with its timestamp, action type, low-level grounding instruction, action rationale, and action parameters. Additionally, we instruct the model to identify the key interface changes resulting from each action execution, along with the underlying reasons for these changes. This annotation strategy is designed to enhance the pretrained model's world model capabilities—given a screenshot and action input, the model can predict the resulting GUI state changes, thereby facilitating long-horizon planning. To enable more fine-grained annotation, we introduce a video segmentation preprocessing step before trajectory extraction. For each input video, the model first segments it into consecutive clips, where each clip corresponds to a single user operation. The prompt template used for annotation is provided in Prompt 3.

For videos exceeding 4 minutes in length, we partition them at 4-minute intervals. When annotating non-initial segments, we provide the model with the annotation results from preceding segments as contextual input, enabling coherent cross-segment annotation. The prompt template for this interactive annotation process is shown in Prompt 4.

## D. Action Spatial Grounding Details

Based on the timestamp $t$, corresponding low-level instruction, and action type obtained from the trajectory extraction phase, we extract three frames from the video at timestamps $t$, $t - 0.5s$, and $t + 0.5s$. We then employ Gemini-3-Pro to perform spatial grounding starting from the first frame. Before outputting the action coordinates, we first prompt the model to determine whether the action can be grounded on the current frame. If grounding is feasible, we adopt the model's grounding result for that frame. Otherwise, the model proceeds to the next frame in the sequence. If all three frames fail to yield a valid grounding result, the corresponding action is discarded from the trajectory. This multi-frame grounding strategy is designed to address temporal misalignment issues that may arise from the high frequency of GUI operations, ensuring robust spatial localization of user actions. The detailed prompt template for our grounding annotation process is provided in Prompt 5.

## E. Statistics

We conduct a statistical analysis of WildGUI, examining the distribution across platforms, applications, and websites. For applications and websites, we categorize them based on their functional purposes. Additionally, we analyze the distribution

*Table 5.* Video Quality Scoring Criteria.

| Dimension | Score | Description |
| --- | --- | --- |
| **Topic Relevance** | 5 (Excellent) | Video is entirely focused on teaching specific software or system operations on target platforms (computers, smartphones, tablets); content is highly feature-oriented, explaining GUI element functions and operation methods rather than completing entire projects; minimal off-topic discussion. |
| | 4 (Good) | Video primarily consists of target platform software instruction, but contains minor supplementary content not directly related to operations (e.g., software history, design philosophy); instruction balances feature demonstration with simple project completion, with core GUI operation segments remaining clearly identifiable. |
| | 3 (Moderate) | Video content has low correlation with software operations or contains substantial non-operational content; may display some GUI operations, but core theme is product reviews, system update news, or other non-tutorial content; or focuses on advanced development or pure theoretical explanation where GUI operations are not the emphasis. |
| | 2 (Low) | Video content is unrelated to software operations or GUI tutorials (entertainment, vlogs, non-educational content); or demonstrates operations on non-target platforms (smart TVs, game consoles, vehicle navigation, IoT devices, industrial equipment, etc.). |
| | 1 (None) | Video content is completely unrelated to GUI operations with no instructional value whatsoever. |
| **Instruction Clarity** | 5 (Very Clear) | Video contains explicit guidance phrases (e.g., "Next, we will..."), provides clear descriptions and explanations for each GUI action (clicking, dragging, typing), and thoroughly explains the rationale or objective behind operations. |
| | 4 (Clear) | Provides basic operational instructions with most GUI steps described and explained, but may lack rationale for operations, or language is not sufficiently precise with minor ambiguities. |
| | 3 (Moderate) | Provides partial operational instructions, but many steps are not clearly described; language is imprecise with heavy colloquialism. |
| | 2 (Low) | Obviously poor instructional quality, filled with colloquial language or excessive off-topic chatter. |
| | 1 (Invalid) | Completely lacks narration or subtitles, or content consists of pure noise/irrelevant information with no extractable instructional value. |
| **Recording Quality** | 5 (High-Quality) | High-definition GUI screen recording with clear and complete visuals; all UI elements and text information are clearly identifiable. |
| | 4 (Cropped) | High-definition GUI screen recording, but the frame is cropped and does not show the complete interface, missing some contextual information. |
| | 3 (Low-Quality) | Contains screen recording but has serious visual issues such as low resolution causing blurred text, obvious compression artifacts, flickering or stuttering, severely affecting recognition. |
| | 2 (External Device) | Content shows screen recording but captured via external devices with unstable footage (shaking, glare, distortion), severely interfering with viewing. |
| | 1 (No Recording) | No actual screen recording (only presenter speaking); or only plays slides without demonstrating actual GUI operations on the target platform. |

*Table 6.* Action Space for Desktop and Mobile Environments.

| Action | Parameters | Description |
|---|---|---|
| ***Desktop Actions*** | | |
| click | x, y | Single click at the specified coordinates. |
| doubleClick | x, y | Double click at the specified coordinates. |
| tripleClick | x, y | Triple click at the specified coordinates. |
| rightClick | x, y | Right click at the specified coordinates. |
| middleClick | x, y | Middle click at the specified coordinates. |
| press | [keys] | Press a single key or a sequence of keys. |
| input | text | Input text into the currently focused element. |
| hotkey | [keys] | Trigger a system hotkey combination. |
| scroll | direction, x, y, distance | Scroll in a direction at specific coordinates. |
| drag | start, end | Drag from start point to end point. |
| moveTo | end_point | Move the cursor to the target position. |
| wait | duration | Pause execution for a given time period. |
| finished | status | End the task and return the goal status. |
| ***Mobile Actions*** | | |
| click | x, y | Tap at the specified coordinates. |
| longpress | x, y, duration | Press and hold for a specific duration. |
| scroll | direction, x, y, distance | Scroll or swipe in the specified direction. |
| pinch | x, y, direction, magpercent | Zoom in or out at the specified coordinates. |
| input | text | Type text into the active input field. |
| drag | start, end | Perform a drag-and-drop gesture. |
| press | [keys] | Simulate hardware keys (Home, Back, etc.). |
| open | app_name | Launch a mobile app by its name. |
| multi_touch | pointers | Execute complex multi-finger gestures. |
| finished | status | Terminate task and report status. |

of video durations, trajectory lengths, and action types. As illustrated in Figure 6 and Figure 7, our dataset encompasses a wide variety of platforms and application categories, ensuring broad coverage and diversity of user tasks across different usage scenarios.

## F. Agent Training

We conduct post-training using a diverse collection of GUI interaction datasets, including Rico (Deka et al., 2017), SeeClickWeb (Cheng et al., 2024), WebUI (Wu et al., 2023), OS-Atlas (Wu et al., 2024), AITW (Rawles et al., 2023), AITZ (Zhang et al., 2024b), AndroidControl (Li et al., 2024), AMEX (Chai et al., 2025), and GUI-Odyssey (Lu et al., 2025c). By leveraging cleaner and more precise human supervision signals, this stage consolidates the agent's policy and enhances performance on domain-specific downstream tasks. The prompt templates used during training are shown in Prompt 6, 7 and 8.

## G. API cost for Video2GUI pipeline

The Video2GUI pipeline supports both closed-source and open-source backbones. We adopt Gemini-3-Pro as our default annotator due to its strong long-context video understanding and spatial grounding quality. We summarize the per-sample API cost when running the full pipeline end-to-end. Trajectory extraction dominates the cost: each sample consumes approximately 15,908 input tokens, 1,338 output tokens, and 1,452 thinking tokens, totalling roughly $0.0653 per sample. Action spatial grounding adds an additional pass on high-resolution frames at approximately $0.011 per sample, while video quality scoring is handled by a self-deployed open-source Qwen2.5-Omni model and incurs negligible cost. The total API cost is therefore approximately $0.0763 per sample. As an open-source alternative we also evaluate Qwen3.5-397B-A17B; while functional, it produces roughly 15–20% lower annotation quality, primarily in temporal alignment and spatial grounding. We therefore adopt Gemini-3-Pro as the default to ensure data quality at scale. Importantly, dataset construction is a one-time cost: we have released WildGUI along with the reproducible Video2GUI pipeline, so downstream users do not

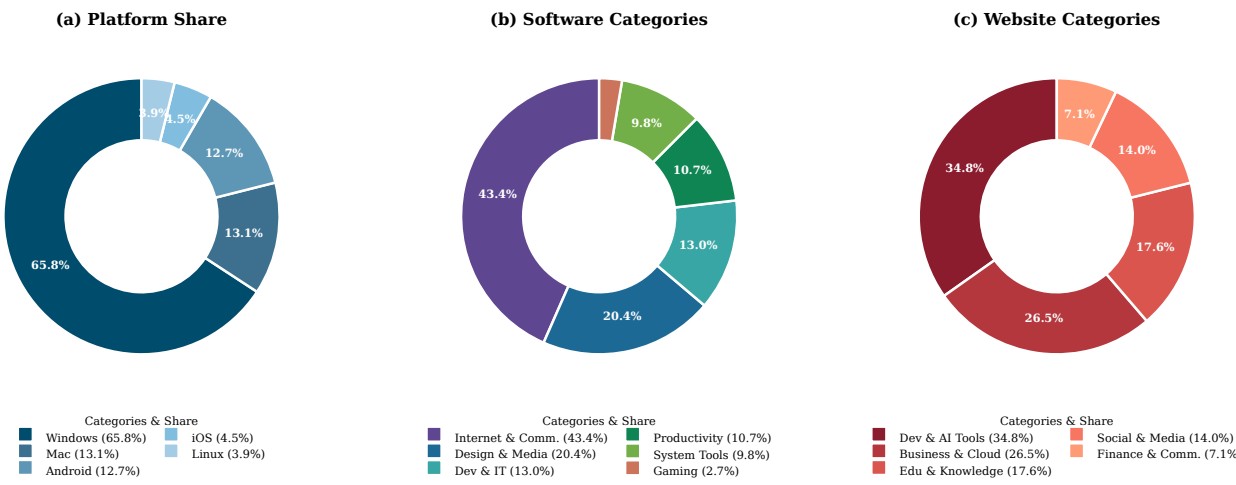

Figure 6. Dataset statistics. Distribution of (a) platforms, (b) software categories, and (c) website categories in WildGUI.

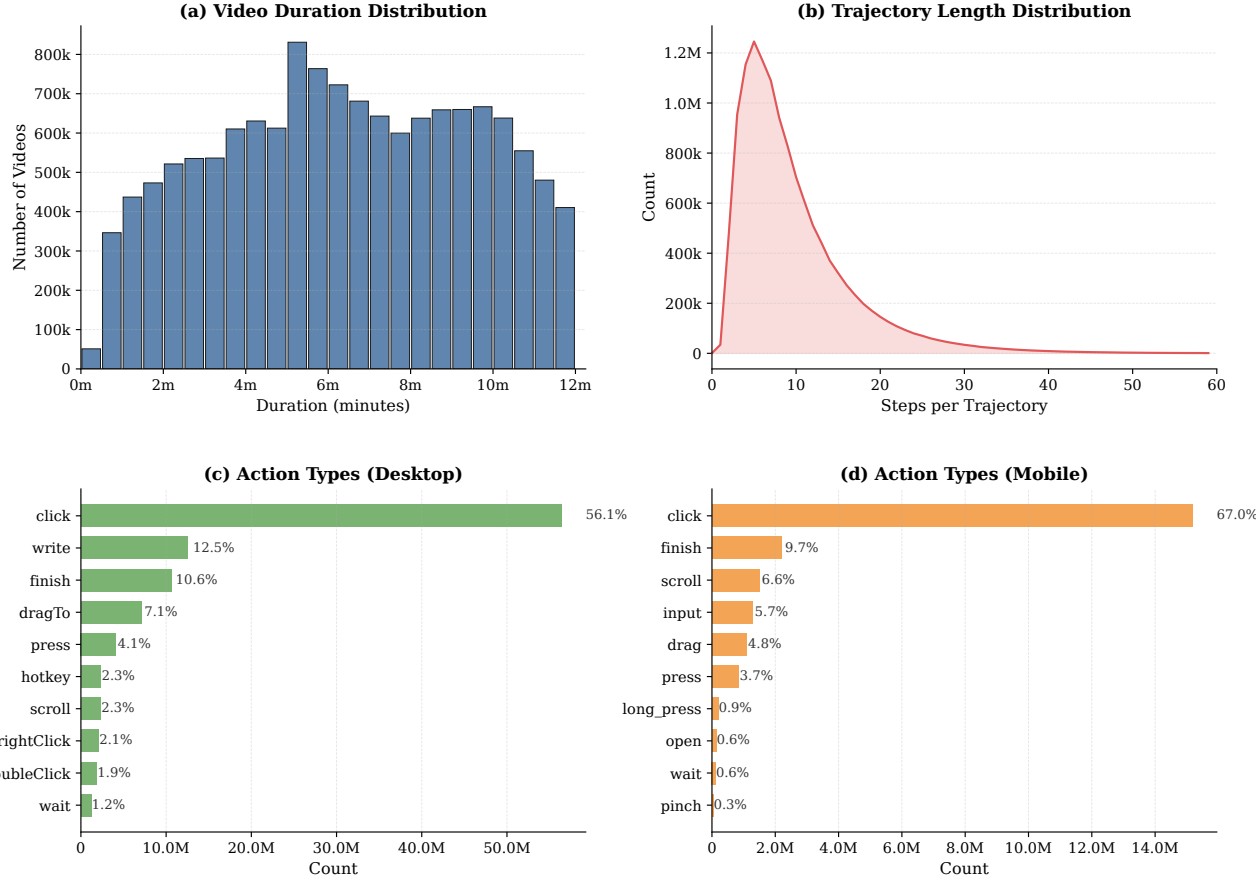

Figure 7. Dataset statistics of WildGUI: (a) video duration distribution; (b) distribution of steps per trajectory; and the action types for (c) desktop and (d) mobile environments.

need to re-run the annotation process.

## H. Evaluation

We evaluate our models across GUI grounding, offline agent evaluation, and online agent evaluation benchmarks to provide a comprehensive assessment of their capabilities.

For GUI grounding, which tests the ability to map natural language instructions to specific UI elements, we utilize OSWorld-G (Xie et al., 2025) and ScreenSpot-Pro (Li et al., 2025). OSWorld-G comprises 564 samples that systematically assess diverse capabilities including text matching, element recognition, layout understanding, and precise manipulation within Linux environments. ScreenSpot-Pro is a challenging benchmark tailored for high-resolution professional scenarios, containing 1,581 expert-annotated tasks across 23 professional applications spanning three operating systems. Its emphasis on industrial software and multi-window interfaces makes it a practical estimate of generalization for GUI visual grounding models.

For offline agent evaluation, we employ AndroidControl (Li et al., 2024) and CAGUI (Zhang et al., 2025) to assess agent capabilities in mobile environments. AndroidControl is categorized into two levels based on instruction granularity: AndroidControl-High evaluates the agent's capability for global reasoning and long-horizon planning, while AndroidControl-Low focuses on fine-grained action execution at the step level. We adopt two standard metrics for evaluation: Type (action type accuracy) and Step-wise Success Rate (SR). Additionally, CAGUI is incorporated to evaluate agentic performance within Chinese-language interfaces.

For online agent evaluation in interactive environments, we use OSWorld (Xie et al., 2024) and AndroidWorld (Rawles et al., 2024). OSWorld is a unified computer environment comprising 369 real-world tasks that span web applications, desktop software, and OS-level operations. AndroidWorld provides a fully functional Android environment with 116 programmatic tasks across 20 real-world apps, featuring dynamic task construction with parameterized natural language instructions.

## I. Examples

We visualize several examples of WildGUI trajectory data, as shown in Figure 8, 9, 10 and 11.

**Prompt 1: Prompt for Meta Info Classification**

```
1   # Role
2   You are an expert data filtering assistant specialized in identifying computer
        usage and software demonstration videos.
3
4   # Task
5   Your task is to analyze the provided **Video Metadata** and determine if the video
        contains **GUI Operation Content**.
6   You must make this judgment solely based on the textual metadata provided. Do not
        attempt to analyze visual frames or audio.
7
8   # Definition: GUI Operation Content
9   A video is considered "Relevant" if its primary content features:
10  1. **Screen Recordings:** Direct capture of a computer, tablet, or smartphone
        screen.
11  2. **Software Interaction:** Users interacting with graphical user interfaces (e.g
        ., clicking, typing, navigating menus, using tools).
12  3. **Application Demos/Tutorials:** Walkthroughs of software (e.g., Photoshop,
        Excel, VS Code, Web Browsers, OS settings).
13
14  A video is considered "Not Relevant" if it is:
15  - Real-world camera footage (Vlogs, IRL, nature, people talking to camera without
        screen share).
16  - Gaming content (unless it is a specific tutorial on UI/Settings).
17  - Static slide presentations (PowerPoint) without active software operation.
18  - Hardware reviews (showing the device physically, not the screen interface).
19
20  # Input Data
21  You will receive the following metadata fields:
22  - Title
23  - Description
24  - Tags/Keywords
25  - Channel Name
26  - Category
27  - Subtitle
28
29  # Output Format
30  Output a valid JSON object with the following fields:
31  - `is_gui_content`: (boolean) true or false.
32  - `confidence`: (float) 0.0 to 1.0.
33  - `reasoning`: (string) A concise explanation citing specific keywords from the
        metadata that led to your decision.
34
35  # Example
36  Input:
37  {
38    "Title": "How to automate data entry in Excel 2024",
39    "Description": "In this video, I show you how to write a Python script and use
        macros.",
40    "Tags": ["productivity", "tutorial", "office", "code"],
41    "Channel": "TechHelp Daily"
42  }
43
44  Output:
45  {
46    "is_gui_content": true,
47    "confidence": 0.95,
48    "reasoning": "Title mentions 'How to' and specific software 'Excel', implying a
        tutorial/walkthrough. Tags include 'tutorial' and 'code'."
49  }
```

**Prompt 2: Prompt for Video Quality Scoring**

```
1  # Role
2
3  You are an expert data filtering assistant specialized in identifying and
       evaluating computer usage and software demonstration videos.
4
5  # Task
6
7  Your task is to analyze the provided **Video Metadata** and assign scores from **1
       to 5** across three specific dimensions. You must make this judgment based on
       the provided textual metadata, descriptions, and transcripts.
8
9  # Evaluation Criteria
10
11 ## 1. Topic Relevance
12
13 **Goal**: Assess if the video focuses on teaching GUI operations on target
       platforms (PC, Smartphone, Tablet) and whether it is "Function-Oriented."
14
15 * **5 (Exceptional)**: Completely focused on target platforms; strictly function-
       oriented; explains GUI elements/tools in depth; zero off-topic content.
16 * **4 (Excellent)**: Focused on target platforms and software/system operations;
       function-oriented; minimal irrelevant talk.
17 * **3 (Good)**: Mainly software teaching but includes minor background (history/
       philosophy); balances function demos with simple projects.
18 * **2 (Moderate)**: Low relevance; focus is on product reviews, news, or high-level
        coding/theory where GUI steps are secondary.
19 * **1 (Low/None)**: Irrelevant content (Vlogs); or operations on non-target
       platforms (TVs, Car systems); no educational value.
20
21 ## 2. Instruction Clarity
22
23 **Goal**: Evaluate the clarity and logic of the verbal or textual instructions
       provided.
24
25 * **5 (Exceptional)**: Professional guiding phrases; every micro-action (click,
       drag, shortcut) is described; provides profound "why" for every step.
26 * **4 (Very Clear)**: Includes clear guiding phrases; every GUI action is
       explicitly described and explained with the "why" behind it.
27 * **3 (Clear)**: Basic instructions for most steps; may lack some "why"
       explanations or contain minor linguistic ambiguity.
28 * **2 (Moderate)**: Partial instructions; many steps are skipped or not described;
       language is imprecise or overly colloquial.
29 * **1 (Poor/None)**: No voiceover/subtitles; pure noise; or instructions are
       dominated by incoherent tangents.
30
31 ## 3. Recording Quality
32
33 **Goal**: Evaluate the visual quality of the screen capture for legibility and
       stability.
34
35 * **5 (Exceptional)**: Ultra-high-definition direct recording; perfect framing;
       every pixel of the UI and text is crystal clear.
36 * **4 (High Quality)**: High-definition direct screen recording; clear and complete
       ; all UI elements are easily legible.
37 * **3 (Cropped High Quality)**: High-definition recording, but the frame is cropped
       , missing some UI context or full-screen interface.
38 * **2 (Low Quality)**: Direct recording with severe issues: low resolution, blurry
       text, compression artifacts, or stuttering.
39 * **1 (Poor/None)**: External recording (camera-to-screen) with shaking/glare; or
       no actual screen recording (PPT/Talking head only).
```

```
40
41 # Output Format
42
43 Please provide your evaluation in the following JSON format:
44
45 ```json
46 {
47   "scores": {
48     "topic_relevance": {
49       "score": 0,
50       "reasoning": "[Briefly explain the 1-5 score regarding platform and function-
            orientation.]"
51     },
52     "instruction_clarity": {
53       "score": 0,
54       "reasoning": "[Briefly explain based on action descriptions and logic.]"
55     },
56     "recording_quality": {
57       "score": 0,
58       "reasoning": "[Briefly explain based on capture method and visual clarity.]"
59     }
60   },
61   "overall_summary": "[A concise summary of the video's instructional value.]"
62 }
63
64 ```
```

**Prompt 3: Prompt for Trajectory Extraction**

```
1 # Role and Objective
2
3 You are an expert, first-person GUI Agent specializing in interaction analysis.
      Your core objective is to analyze a GUI interaction video and produce a two-
      part analysis. You must strictly follow the output format specified, generating
       the "Shot Splitting" section first, followed by the "Task Annotations" JSON
      list.
4
5 Critical Style Instruction: When generating annotations, do not act as a passive
      observer. Act as the Agent performing the task. Your reasoning must reflect a
      continuous thought process, linking past history to current decisions.
6
7 Your output must be in two distinct parts, in this exact order:
8 Part 1: Shot Splitting (Markdown)
9 Part 2: Task Annotations (A single JSON list)
10
11 ## Part 1: Shot Splitting
12 First, you must segment the entire video into granular "shots" and output them as a
      numbered markdown list.
13 Shot Definition: A shot is the most fundamental unit of the video, defined by a
      start time and an end time.
14 The start_time (mm:ss) is the precise moment a user's atomic action (e.g., click,
      key press, scroll) begins.
15 The end_time (mm:ss) is the start_time of the very next atomic action.
16 Therefore, a shot [start_time_1 - end_time_1] covers the action at start_time_1 and
       the entire resulting screen state until the next action begins at start_time_2
       (which is the same as end_time_1).
17
18 **Output Format (Part 1):**
```

```
19  You must output this section exactly as follows, starting with the header:
20  Shot Splitting
21  1. mm:ss - mm:ss
22  2. mm:ss - mm:ss
23  3. mm:ss - mm:ss
24  ...
25  (Split as granularly as possible.)
26
27  You must strictly confine your analysis to the provided video segment ONLY. Do NOT
        generate any content for timestamps beyond the end of this provided segment.
28
29  ## Part 2: Task Annotations
30  Second, after the "Shot Splitting" section, you must output a single JSON list
        [{...}, {...}].
31  Each item {} in this list represents one complete, high-level sub-task.
32  A single sub-task will contain several of the atomic shots identified in Part 1.
33  You must not output any text or explanations before or after this JSON list.
34  Every completed sub-task must conclude with a finish action representing the
        completion of the task.
35
36  **Output Format (Part 2):**
37  The JSON list must strictly adhere to the following structure.
38  ```json
39  [
40      {
41          "task_id": 0,
42          "instruction": "A one-sentence summary of the user's core intent for this
                task.",
43          "dense_caption": "A detailed, causal description of this task.",
44          "plan": "The high-level steps for this task, formatted as 'step1: ...,
                step2: ...'.",
45          "platform": "The operating system (e.g., 'windows', 'ios', 'android').",
46          "software": "The name of the primary application (e.g., 'WeChat', 'Google
                Chrome').",
47          "website": "The website domain if a browser is used (e.g., 'youtube.com'),
                else null.",
48          "user_actions": [
49              {
50                  "timestamp": "Timestamp (mm:ss) from Part 1, marking the start of
                        the shot.",
51                  "action_type": "The type of action (e.g., 'click', 'write', 'scroll
                        ').",
52                  "grounding_instruction": "A natural language instruction to
                        precisely locate the UI element for this action.",
53                  "action_reason": "First-person reasoning. Synthesize previous
                        trajectory + current state => next action.",
54                  "action_parameters": {
55                      "...": "Parameters specific to the action_type, defined by the
                            platform."
56                  },
57                  "core_change_reason": "Detailed explanation of the system logic/
                        mechanism behind the screen change.",
58                  "core_change": "The most critical and direct on-screen change
                        resulting from the action (appearance, text updates, modal
                        closing, etc.)."
59              }
60          ]
61      }
62  ]
63  ```
64
```

```
65  **Field Definitions for JSON (Part 2)**
66  task_id (Integer): A unique identifier for the sub-task, starting from 0.
67  instruction (String): A single-sentence summary of the user's core intent for this
        sub-task.
68  dense_caption (String): A detailed description that includes causality for this sub
        -task.
69  plan (String): The high-level steps to complete this sub-task.
70  platform (String): The operating system. Must be one of: windows, mac, android, ios
        , linux. This choice determines which Action Space (PC or Mobile) is used for
        the user_actions.
71  software (String): The name of the primary application being used (e.g., 'Google
        Chrome', 'WeChat', 'Photoshop').
72  website (String or null): If the software is a web browser, specify the primary
        website domain (e.g., 'youtube.com', 'google.com'). Must be null if not a web
        task.
73  user_actions (Array): A chronologically ordered list of user_action objects. Each
        object corresponds to one shot from Part 1. The structure of action_type and
        action_parameters must match the Action Space for the platform specified above.
74  timestamp (String): The start_time (mm:ss) of the shot from Part 1.
75  action_type (String): The type of action performed. Must be one of the types from
        the correct platform-specific Action Space below.
76  grounding_instruction (String): A clear, concise natural language instruction that
        uniquely identifies the UI element being interacted with. For example: "Click
        the 'File' menu in the top-left corner," or "Tap the input field labeled '
        Username'."
77  action_reason (String): Do not simply state "To do X." Instead, simulate the Agent'
        s thought process. You must explicitly synthesize the trajectory history (what
        I just did) with the current observation (what I see now) to derive the next
        step. Bad: "To open the file." Good: "I have successfully navigated to the
        target folder. Now that I see the file list, I need to select the specific file
        'report.pdf' to proceed with the upload."
78  action_parameters (Object): Parameters for the action. The structure of this object
        is strictly defined by the action_type and the platform, as specified in the "
        Platform-Specific Action Spaces" section.
79  core_change_reason (String): A detailed technical or logical explanation of the
        system's mechanism. Explain why the screen is about to change based on software
        design patterns. Example: "Clicking the 'Open' button in a system file dialog
        triggers the OS to validate the file path, close the modal window, and return
        the file handle to the web browser."
80  core_change (String): Based on your concrete evidence and valid reasoning, provide
        a detailed description of the most likely predictable content of the next shot.
        Your prediction must be the necessary outcome of your reasoning process, with
        every detail being justifiable.
81
82  **Platform-Specific Action Spaces**
83  You must use one of the following action spaces based on the task's platform. The
        action_type string determines the exact structure of the action_parameters
        object. Coordinates [y, x] are in pixels.
84  A. PC Action Space (for platform: 'windows', 'mac', 'linux' etc.)
85  action_type: "click"
86  action_parameters: { "point": "[y, x]", "bbox": "[y1, x1, y2, x2]" }
87  action_type: "doubleClick"
88  action_parameters: { "point": "[y, x]", "bbox": "[y1, x1, y2, x2]" }
89  action_type: "tripleClick"
90  action_parameters: { "point": "[y, x]", "bbox": "[y1, x1, y2, x2]" }
91  action_type: "rightClick"
92  action_parameters: { "point": "[y, x]", "bbox": "[y1, x1, y2, x2]" }
93  action_type: "middleClick"
94  action_parameters: { "point": "[y, x]", "bbox": "[y1, x1, y2, x2]" }
95  action_type: "press" (for keyboard keys)
96  action_parameters: { "key_name": "e.g., 'enter', 'esc', 'delete', 'tab', 'F5'" }
```

```
97  action_type: "write" (for typing text)
98  action_parameters: { "text": "The text content to be typed" }
99  action_type: "hotkey"
100 action_parameters: { "keys": "e.g., 'Ctrl+C', 'Alt+F4', 'Cmd+S'" }
101 action_type: "scroll" (vertical)
102 action_parameters: { "direction": "[up, down]", "point": "[y, x]", "
        magnitude_pixels": "e.g., 100" }
103 action_type: "hscroll" (horizontal)
104 action_parameters: { "direction": "[left, right]", "point": "[y, x]", "
        magnitude_pixels": "e.g., 100" }
105 action_type: "moveTo"
106 action_parameters: { "point": "[y, x]" }
107 action_type: "dragTo"
108 action_parameters: { "start_point": "[y, x]", "end_point": "[y, x]" }
109 action_type: "wait"
110 action_parameters: { "duration": "e.g., 100" } (in milliseconds)
111 action_type: "finish"
112 action_parameters: { "status": "success" }
113
114 B. Mobile Action Space (for platform: 'ios', 'android' etc.)
115 action_type: "click" (tap)
116 action_parameters: { "point": "[y, x]", "bbox": "[y1, x1, y2, x2]" }
117 action_type: "long_press"
118 action_parameters: { "point": "[y, x]", "duration_ms": 500, "bbox": "[y1, x1, y2,
        x2]" }
119 action_type: "scroll"
120 action_parameters: { "direction": "[down|up|left|right]" }
121 action_type: "pinch"
122 action_parameters: { "center_point": "[y, x]", "direction": "[in, out]", "
        magnitude_percent": "e.g., 30"}
123 action_type: "input" (for text entry)
124 action_parameters: { "text": "The text content" }
125 action_type: "drag"
126 action_parameters: { "start_point": "[y, x]", "end_point": "[y, x]", "duration_ms":
        "e.g., 1000" }
127 action_type: "press" (for system/hardware keys)
128 action_parameters: { "key": "[back, home, enter, volume_up, volume_down, power...]"
        }
129 action_type: "open"
130 action_parameters: { "app": "Name of app to open" }
131 action_type: "multi_touch_gesture"
132 action_parameters: { "pointers": [ { "id": 0, "path": "[[y1, x1], [y2, x2], ...]"
        }, { "id": 1, "path": "[[y3, x3], [y4, x4], ...]" } ] }
133 action_type: "finish"
134 action_parameters: { "status": "success" }
135
136 **Try best efforts, response at least 5k tokens.**
```

### Prompt 4: Prompt for Iterative Annotation

```
1  # CONTEXT
2
3  You are performing a continuous, shot-by-shot analysis of a single video, segment
       by segment.
4  You have already analyzed the video up to **[Current Start Time]**. Your complete
       analysis so far is provided below as "Annotation History".
5
6  # ANNOTATION HISTORY (Context from 00:00 to **[Current Start Time]**)
```

```
 7
 8  # ===
 9  **[Insert Previous Analysis History Here]**
10
11  # CURRENT TASK
12
13  You have been provided with the video clip from **[Current Start Time]** to **[
      Current End Time]**. You must now analyze this specific segment.
14
15  **CRITICAL INSTRUCTIONS:** You must strictly follow the following rules:
16
17  1. **Use Absolute Timestamps:** All timestamps in your output (e.g., "Shot
      Segmentation") must be **absolute** (relative to the *full* video), not
      relative to the current clip.
18  2. **Maintain Consistency:** Your new analysis must be fully **consistent** with
      the "Annotation History". Ensure that your interpretations, character analyses,
       and narrative arcs logically continue from the previous annotations.
19  3. **Use Global Context:** You must use the "Annotation History" to understand the
      **global context** (e.g., established narrative, characters, themes) to better
      interpret the events within the current clip. You are encouraged to cite
      specific evidence from the history (e.g., "which mirrors the action at 04:30")
      to enhance your interpretation.
20  4. **Do Not Mention Text in History:** The "Annotation History" text is provided *
      only* for your internal context. Your output must **never** mention the
      existence of this text (e.g., do not write "Based on the history provided..."
      or "In the previous annotation..."). Your analysis must read as a single,
      seamless continuation of the history, as if you were analyzing the entire video
       in one go.
21  5. **Continue unfinished tasks:** If the last sub-task in the preceding part is
      incomplete, retain its task index and finish every remaining clip within that
      sub-task. If the last sub-task is already complete, start a fresh sub-task with
       the next available index.
22
23  A more detailed task instruction for the current **[Current Start Time] – [Current
      End Time]** segment is as follows:
24
25  **[Insert Specific Task Prompt Here]**
```

**Prompt 5: Prompt for Action Spatial Grounding**

```
 1  You are a GUI agent.
 2
 3  You are given the natural language `grounding_instruction` to locate, the `
      action_type` and the `screenshot`.
 4
 5  You need to perform the following two steps:
 6
 7  1.  **Feasibility Check:**
 8
 9      * Analyze the `screenshot` and `grounding_instruction`.
10      * Determine if the target element mentioned in the instruction (e.g., button,
          text field, icon) is **actually present and visible** in the current `
          screenshot`.
11
12  2.  **Grounding & Prediction:**
13
14      * **If not feasible** (e.g., the 'Submit' button mentioned in the instruction
          does not exist in the screenshot), set `feasible` to `false` in the
```

```
       output and provide a reason.
15     * **If feasible** (the target element is visible), set `feasible` to `true`.
         Then, for **each** point in the `points_to_predict` list (e.g., `point`),
          you must predict its precise center point and the bounding box of the
         target element it corresponds to.
16
17  -----
18
19  ## Inputs
20
21  ### 1\. Grounding Instruction
22
23  `{grounding_instruction}`
24
25  ### 2\. Action Type
26
27  `{action_type}`
28
29
30  ## Output Format
31
32  You must **strictly** provide your analysis and predictions in the following JSON
       format.
33
34  > **Note:** The bounding boxes and center points must use relative coordinates
       (0-1000) and be formatted as follows: `<bbox>y1 x1 y2 x2</bbox>` and `<point>y
       x</point>`.
35
36  ### Example: Action is Feasible
37
38  ```json
39  {
40    "feasible": true,
41    "predictions": [
42      {
43        "point_name": "start_point",
44        "center_point": "<point>450 320</point>",
45        "bounding_box": "<bbox>400 300 500 340</bbox>"
46      },
47      {
48        "point_name": "end_point",
49        "center_point": "<point>450 320</point>",
50        "bounding_box": "<bbox>400 300 500 340</bbox>"
51      }
52    ]
53  }
54  ```
55
56  The point_name you output must exactly match the parameter name defined in the `
       Action Type` section.
57
58  ### Example: Action is Not Feasible
59
60  ```json
61  {
62    "feasible": false,
63    "reason": "The target mentioned in the instruction (e.g., 'Submit button') was
         not found in the current screenshot."
64  }
65  ```
66
```

```
67  **Remember:**
68  1. The center points must use relative coordinates (0-1000) and be formatted as
        follows: `<point>y x</point>`.
69  2. The bounding boxes must use relative coordinates (0-1000) and be formatted as
        follows: `<bbox>y1 x1 y2 x2</bbox>`.
```

**Prompt 6: Prompt for GUI Grounding**

```
1  1. Locate UI components that match the command: \"{}\". Output a JSON in the format
       [{{\"point\": [...], \"label\": \"{{the_whole_command}}\"}}, ...].
2
3  2. Locate UI components that match the command: \"{}\". Output a JSON in the format
       [{{\"bbox_2d\": [...], \"label\": \"{{the_whole_command}}\"}}, ...].
```

**Prompt 7: Prompt for GUI Action Prediction**

```
1  1. You are a GUI agent. You will be provided with a screenshot, a goal, and your
       action history. You need to perform the next action to complete the task.\n\n##
        Action Space\n{}\n\n## Goal\n{}\n\n## Previous Actions\n{}\n\nNow, output the
       next action in json format [{{\"action\": \"{{action_name}}\"}}, ...].
2
3  2. You are a GUI agent. You will be provided with a screenshot, a goal, and your
       action history. You need to analyze the current situation and perform the next
       action to complete the task.\n\n## Action Space\n{}\n\n## Goal\n{}\n\n##
       Previous Actions\n{}\n\nPlease output your response in the following format:\n\
       nThought: <your reasoning about what to do next>\nAction: <the action you will
       take>\n\n```json\n[{{\"action\": \"{{action_name}}\"}}, ...]\n```
```

**Prompt 8: Prompt for GUI Trajectory Modeling**

```
1  1. Based on the screenshot and the goal: \"{}\", perform the next action to
       complete the task. Output a JSON in the format [{{\"action\": \"{{action_name
       }}\"}}, ...].
2
3  2. Based on the screenshot and the goal: \"{}\", analyze the current situation and
       perform the next action to complete the task.\n\nPlease output your response in
        the following format:\n\nThought: <your reasoning about what to do next>\
       nAction: <the action you will take>\n\n```json\n[{{\"action\": \"{{action_name
       }}\"}}, ...]\n```
```

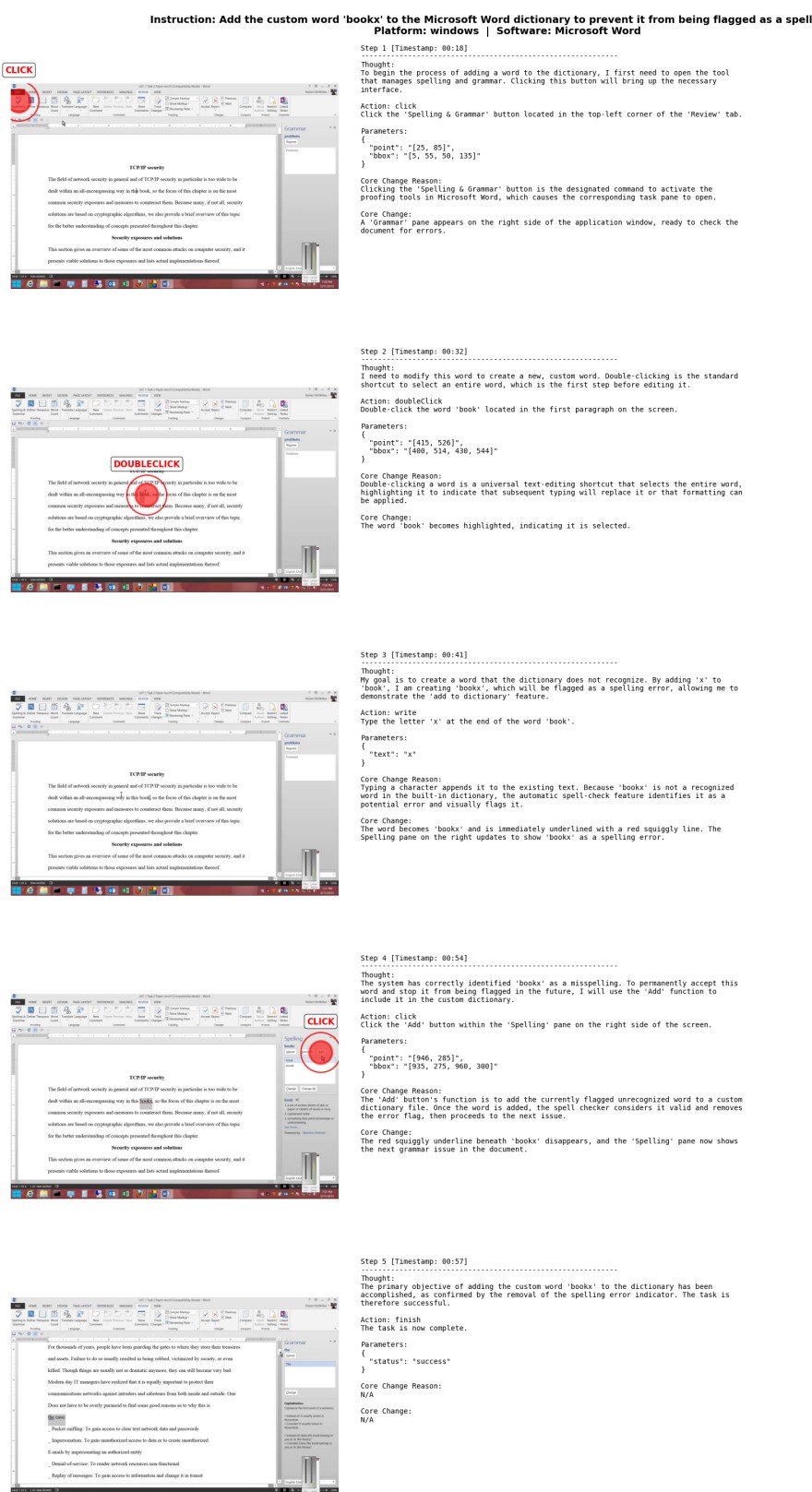

*Figure 8.* WildGUI example.

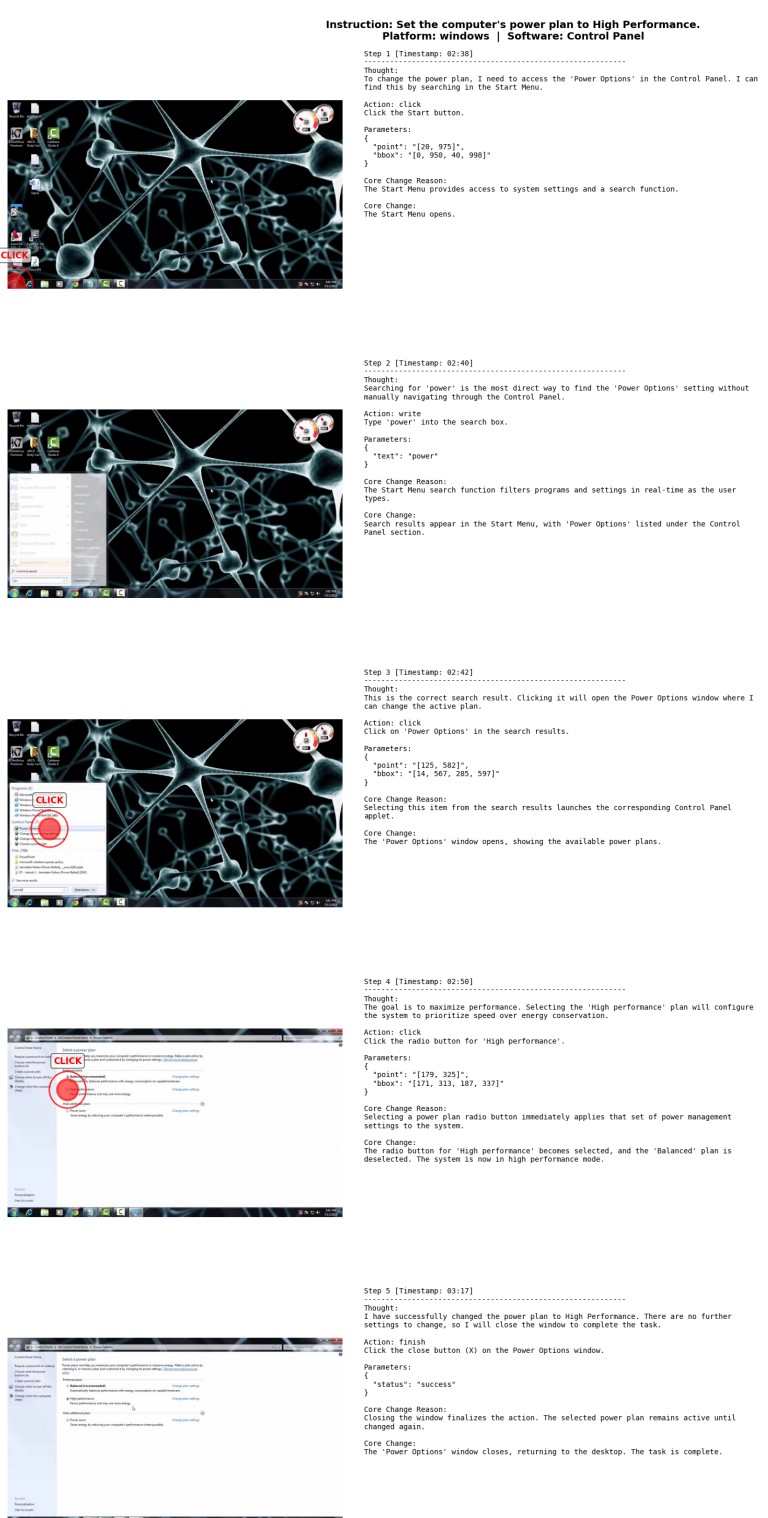

*Figure 9.* WildGUI example.

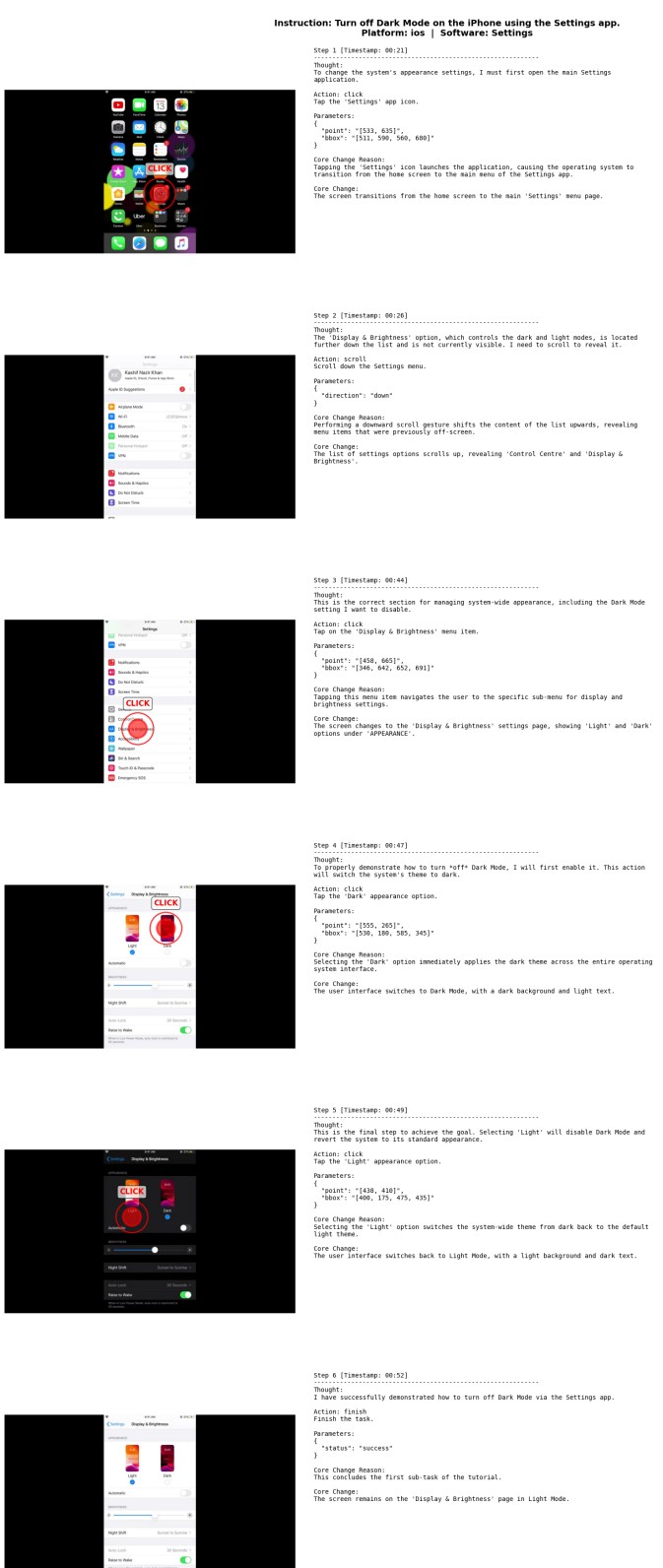

*Figure 10.* WildGUI example.

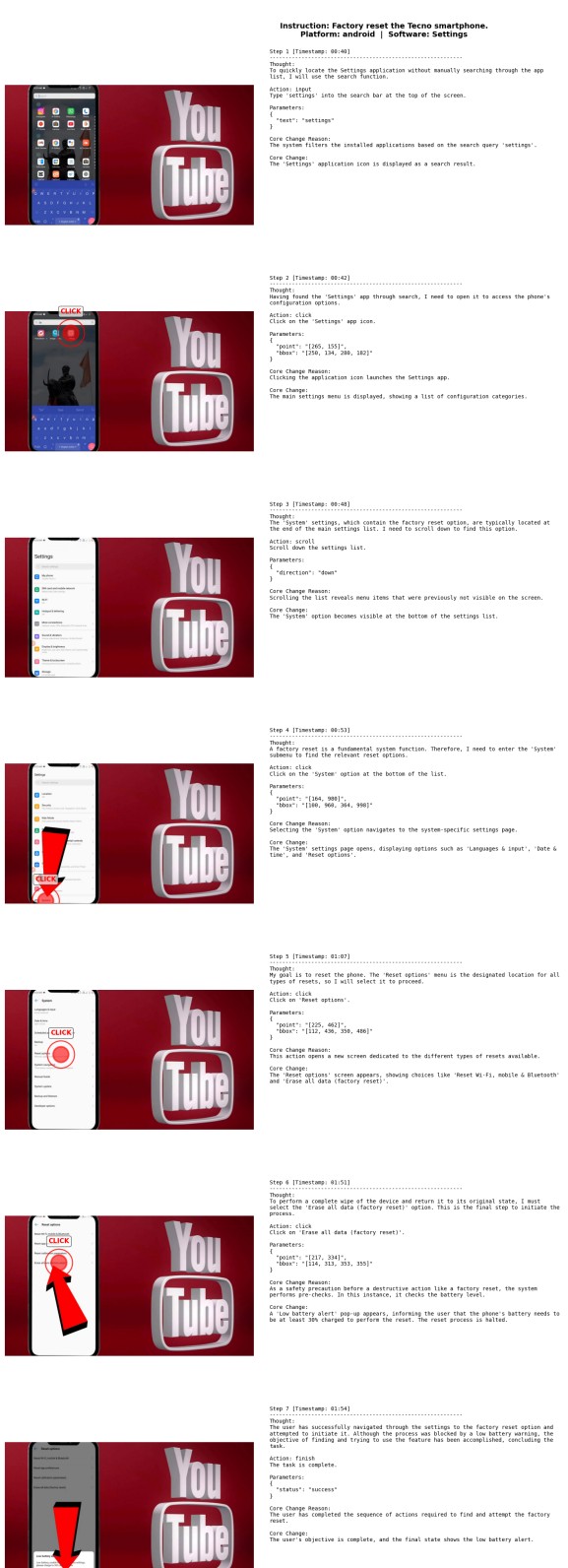

**Instruction: Factory reset the Tecno smartphone.**
**Platform: android | Software: Settings**

Step 1 [Timestamp: 00:40]
-----------------------------------------------------------
Thought:
To quickly locate the Settings application without manually searching through the app
list, I will use the search function.

Action: input
Type 'settings' into the search bar at the top of the screen.

Parameters:
{
  "text": "settings"
}

Core Change Reason:
The system filters the installed applications based on the search query 'settings'.

Core Change:
The 'Settings' application icon is displayed as a search result.

Step 2 [Timestamp: 00:42]
-----------------------------------------------------------
Thought:
Having found the 'Settings' app through search, I need to open it to access the phone's
configuration options.

Action: click
Click on the 'Settings' app icon.

Parameters:
{
  "point": "[265, 155]",
  "bbox": "[250, 134, 280, 182]"
}

Core Change Reason:
Clicking the application icon launches the Settings app.

Core Change:
The main settings menu is displayed, showing a list of configuration categories.

Step 3 [Timestamp: 00:48]
-----------------------------------------------------------
Thought:
The 'System' settings, which contain the factory reset option, are typically located at
the end of the main settings list. I need to scroll down to find this option.

Action: scroll
Scroll down the settings list.

Parameters:
{
  "direction": "down"
}

Core Change Reason:
Scrolling the list reveals menu items that were previously not visible on the screen.

Core Change:
The 'System' option becomes visible at the bottom of the settings list.

Step 4 [Timestamp: 00:53]
-----------------------------------------------------------
Thought:
A factory reset is a fundamental system function. Therefore, I need to enter the 'System'
submenu to find the relevant reset options.

Action: click
Click on the 'System' option at the bottom of the list.

Parameters:
{
  "point": "[184, 980]",
  "bbox": "[100, 960, 364, 990]"
}

Core Change Reason:
Selecting the 'System' option navigates to the system-specific settings page.

Core Change:
The 'System' settings page opens, displaying options such as 'Languages & input', 'Date &
time', and 'Reset options'.

Step 5 [Timestamp: 01:07]
-----------------------------------------------------------
Thought:
My goal is to reset the phone. The 'Reset options' menu is the designated location for all
types of resets, so I will select it to proceed.

Action: click
Click on 'Reset options'.

Parameters:
{
  "point": "[225, 462]",
  "bbox": "[112, 438, 350, 486]"
}

Core Change Reason:
This action opens a new screen dedicated to the different types of resets available.

Core Change:
The 'Reset options' screen appears, showing choices like 'Reset Wi-Fi, mobile & Bluetooth'
and 'Erase all data (factory reset)'.

Step 6 [Timestamp: 01:51]
-----------------------------------------------------------
Thought:
To perform a complete wipe of the device and return it to its original state, I must
select the 'Erase all data (factory reset)' option. This is the final step to initiate the
process.

Action: click
Click on 'Erase all data (factory reset)'.

Parameters:
{
  "point": "[217, 334]",
  "bbox": "[114, 313, 353, 355]"
}

Core Change Reason:
As a safety precaution before a destructive action like a factory reset, the system
performs pre-checks. In this instance, it checks the battery level.

Core Change:
A 'Low battery alert' pop-up appears, informing the user that the phone's battery needs to
be at least 30% charged to perform the reset. The reset process is halted.

Step 7 [Timestamp: 01:54]
-----------------------------------------------------------
Thought:
The user has successfully navigated through the settings to the factory reset option and
attempted to initiate it. Although the process was blocked by a low battery warning, the
objective of finding and trying to use the feature has been accomplished, concluding the
task.

Action: finish
The task is complete.

Parameters:
{
  "status": "success"
}

Core Change Reason:
The user has completed the sequence of actions required to find and attempt the factory
reset.

Core Change:
The user's objective is complete, and the final state shows the low battery alert.

*Figure 11.* WildGUI example.

