# OpenReview forum: "Video2GUI: Synthesizing Large-Scale Interaction Trajectories for Generalized GUI Agent Pretraining"
_ICML.cc/2026/Conference — ICML 2026 regular_

### Official Review · Reviewer_zvnt · 2026-03-11

**Soundness:** 3
**Presentation:** 4
**Significance:** 3
**Originality:** 4
**Overall Recommendation:** 4
**Confidence:** 5

**Summary:**

This paper presents Video2GUI, an automated framework designed to synthesize large-scale, grounded GUI interaction trajectories from unlabeled web videos. To address the data scarcity challenge in training generalized GUI agents, the authors introduce a "coarse-to-fine" filtering strategy to identify high-quality tutorials from a massive pool of 500 million videos. The framework leverages advanced multimodal large language models (specifically Gemini-3-Pro) to automatically annotate task instructions, action timestamps, and precise spatial coordinates. Based on this pipeline, the authors curate the WildGUI dataset, which contains 12.7 million trajectories covering over 1,500 diverse real-world applications. Experimental results demonstrate that continuous pre-training on WildGUI significantly boosts the performance of models such as Qwen2.5-VL and Mimo-VL on established benchmarks like ScreenSpot-Pro and OSWorld.

**Compliance With Llm Reviewing Policy:**

Affirmed.

**Final Justification:**

I will maintain my positive evaluation and look forward to the final version.

**Key Questions For Authors:**

See Weaknesses and Questions

**Limitations:**

yes

**Strengths And Weaknesses:**

Strengths:

Significant Practical Value: The framework synthesizes high-quality interaction trajectories from unlabeled internet videos, addressing a critical data scarcity bottleneck in the GUI Agent field.

Scalability and Diversity: The resulting WildGUI dataset is impressive in scale, featuring over 12 million trajectories across a wide variety of 1,500+ real-world applications and websites.

Solid Performance Gains: The reported 5-20% improvement across multiple GUI grounding and action benchmarks demonstrates the effectiveness of the proposed pre-training data.

Weaknesses and Questions:

Ambiguity in Data Sources: While the paper mentions collecting metadata from "public repositories and large-scale web crawling" (Section 3.1), it does not explicitly specify which video platforms (e.g., YouTube, Bilibili, etc.) were utilized. Clarifying the sources is essential for understanding potential data biases and ensuring reproducibility.

Verification of Localization Accuracy: The paper relies on Gemini-3-Pro for action spatial grounding and claims a 95% accuracy based on a manual check of 200 samples. Given the critical nature of coordinate precision for GUI agents, more extensive quantitative data or a detailed breakdown (e.g., performance across different action types or resolutions) is needed to justify if Gemini-3-Pro’s localization is consistently reliable at this scale.

Persuasiveness of Baselines (OSWorld): In the OSWorld experiments, the authors primarily showcase improvements on the Mimo-VL-7B baseline. However, Mimo-VL is known to have relatively weak native GUI capabilities. To provide a more convincing demonstration of WildGUI's value, it is recommended to evaluate its impact on a stronger, state-of-the-art open-source baseline, such as OpenCUA-7B or the latest Qwen2-VL models, as shown in Table 2. Showing that the dataset can further enhance already strong models would better validate its contribution to the community.

Overall, I recognize the significant contribution of this work and look forward to the open-sourcing of the dataset and models, which will undoubtedly foster further development and innovation within the GUI agent research community.

---

> ### Author Rebuttal · Authors · 2026-03-30
>
> We thank Reviewer zvnt for the positive assessment and constructive suggestions.
>
> ---
>
> > ## Q1: Ambiguity in data sources
>
> Thanks for the suggestion. We clarify that our video metadata is collected from YouTube, which is mentioned in Section 6.2 (Data Collection for GUI Agents) of our paper. We chose YouTube as the primary data source because it offers excellent diversity, covering a wide range of software tutorials across multiple languages, countries, and platforms. This global coverage naturally provides cross-lingual and cross-cultural GUI interaction data, as evidenced by our great improvements on the Chinese-language CAGUI benchmark. We will explain the data source more explicitly in the revised paper. Additionally, we will open-source the collected metadata to ensure full reproducibility of our pipeline. The detailed distribution analysis of platforms, application categories, and website categories has been provided in Appendix E (Statistics) and visualized in Figure 6 and Figure 7.
>
> > ## Q2: Verification of localization accuracy
>
> We agree that more extensive verification is needed. We have expanded our evaluation to 600 manually checked samples across three platforms (Website, Mobile, Desktop) with 200 samples each. We further analyze grounding accuracy across five major action types, with 120 samples per action type. The results are summarized below:
>
> | Action Type | Samples | Accuracy |
> |-------------|---------|----------|
> | Click       | 120     | 96.7%    |
> | Scroll      | 120     | 93.3%    |
> | Input/Write | 120     | 91.7%    |
> | Drag/DragTo | 120     | 94.2%    |
> | Press       | 120     | 93.3%    |
> | Overall     | 600     | 93.8%    |
>
> Furthermore, the significant improvements on ScreenSpot-Pro (+15.1 for Qwen2.5-VL, +15.7 for Mimo-VL) and OSWorld-G (+26.4 for Qwen2.5-VL, +12.9 for Mimo-VL) after continual pre-training on WildGUI also demonstrate that the localization quality is reliable at this scale, as these grounding benchmarks directly evaluate the model's ability to precisely locate UI elements. We will include this expanded evaluation in the revised paper.
>
> > ## Q3: Evaluation of stronger baselines
>
> We appreciate the suggestion. We have conducted additional experiments on a stronger baseline. After continual pre-training on WildGUI, Qwen3-VL-8B-Instruct improves from 33.9% to 37.9% on OSWorld, demonstrating that WildGUI can further enhance models with already strong GUI capabilities. This confirms that the benefit of WildGUI continual pre-training is not limited to weaker models but generalizes to state-of-the-art architectures. We will include these results in the revised paper.
>
> ---
>
> We are committed to open-sourcing both the WildGUI dataset and the Video2GUI pipeline to support the research community, and we appreciate the reviewer's recognition of this contribution's value.

---

> > ### Author Rebuttal · Reviewer_zvnt · 2026-04-03
> >
> > I would like to thank the authors for their thorough and high-quality rebuttal.
> >
> > I am satisfied with these responses and will maintain my positive evaluation. I look forward to seeing all these revisions and additional results incorporated into the final version of the manuscript.

---

### Official Review · Reviewer_XfPh · 2026-03-12

**Soundness:** 3
**Presentation:** 3
**Significance:** 3
**Originality:** 2
**Overall Recommendation:** 5
**Confidence:** 4

**Summary:**

This paper presents Video2GUI, a fully automated framework for extracting grounded GUI interaction trajectories from unlabeled internet videos. The pipeline consists of three stages: (A) coarse-to-fine video filtering using metadata-based textual filtering followed by a visual quality scoring model, (B) trajectory extraction using Gemini-3-Pro with a sliding-window strategy to convert video segments into instruction-trajectory sequences, and (C) action spatial grounding that maps extracted actions to precise screen coordinates. Applying this pipeline to 500 million video metadata entries, the authors construct WildGUI, a large-scale dataset containing 12.7 million interaction trajectories spanning over 1,500 applications and websites with 124.5 million screenshots. WildGUI covers web, mobile, and desktop platforms with both high-level and low-level instructions. The authors propose a two-stage agent training strategy combining continual pre-training on WildGUI with supervised post-training. Experiments on Qwen2.5-VL-7B and Mimo-VL-7B demonstrate consistent improvements of 5-20% across multiple GUI grounding (ScreenSpot-Pro, OSWorld-G) and action benchmarks (AndroidControl, CAGUI), matching or surpassing state-of-the-art performance. Scaling analysis shows a positive correlation between data quantity and performance, and ablation studies validate the contribution of each pipeline component and training objective.

**Compliance With Llm Reviewing Policy:**

Affirmed.

**Final Justification:**

After reviewing the authors' rebuttal, I raise my score to 5. While the limited scope remains a concern, it does not substantially diminish the paper's contribution as a solid accepted paper.

**Key Questions For Authors:**

1. How does the pipeline handle the distribution shift between YouTube tutorial videos (which tend to be curated, well-lit, and narrated) and real-world GUI usage scenarios that may involve more diverse and noisy interaction patterns? Have the authors analyzed whether the pretraining gains transfer to non-tutorial-style tasks?

2. What is the estimated total API cost for running the full Video2GUI pipeline (Gemini-3-Pro for quality scoring, trajectory extraction, and grounding on 4.2M videos)? This information is important for assessing the practical reproducibility of the approach.

3. Could the authors provide a more detailed breakdown of grounding accuracy across different platforms (web, mobile, desktop) and action types (click, type, scroll, drag)? The current 95% accuracy from 200 random samples is insufficient to characterize quality across the heterogeneous dataset.

4. How does WildGUI compare to VideoAgentTrek (Lu et al., 2025) in terms of trajectory quality and downstream performance? A direct comparison would help contextualize the contribution.

5. The paper mentions using Qwen2.5-7B for coarse metadata filtering. How sensitive is the final dataset quality to the choice of this filtering model? Have the authors experimented with alternative models or thresholds?

**Limitations:**

Yes.

**Strengths And Weaknesses:**

Strengths:

1. Scale of the dataset: WildGUI is a large-scale dataset (12.7M trajectories, 124.5M images) spanning web, mobile, and desktop platforms. The sheer volume of data is a practical contribution, though scale alone does not constitute a scientific advance.

2. Reasonable pipeline design: The coarse-to-fine video filtering and sliding-window trajectory extraction are sensible engineering choices, though they rely heavily on off-the-shelf models (Gemini-3-Pro, DeepSeek-V3) without meaningful methodological innovation.

Weaknesses:

1. Reproducibility concerns and lack of methodological novelty: The entire pipeline is critically dependent on proprietary, closed-source models (Gemini-3-Pro for video quality scoring and trajectory extraction, DeepSeek-V3 for metadata filtering). This is a fundamental flaw—the core contribution (the dataset) cannot be independently verified or reproduced. Similar pipelines have been explored in AgentTrek (Xu et al., 2024) and VideoAgentTrek (Lu et al., arXiv:2510.19488, 2025), which also use VLMs for trajectory extraction. The individual components (metadata filtering, VLM-based annotation, spatial grounding) are straightforward applications of existing techniques with no novel algorithmic contribution.

2. Questionable data quality with insufficient validation: The paper claims 95% grounding accuracy from a manual review of only 200 randomly sampled actions—this is a woefully inadequate sample size for a dataset of 12.7M trajectories. No stratified analysis across platforms, applications, or action types is provided. Error modes and failure cases are not characterized. For a dataset paper, the quality assurance methodology is far below the standard expected at a top venue.

3. Incomplete and unfair experimental comparisons: Table 2 omits several strong recent baselines including SeeClick (Cheng et al., 2024) on the grounding task, and more recent proprietary models on action benchmarks. Critically, the paper does not compare against VideoAgentTrek (Lu et al., 2025), which directly addresses the same problem of mining GUI trajectories from unlabeled videos. The online evaluation results are also underwhelming: the improvements on OSWorld (12.3% SR) are modest compared to the Stage 2 Only baseline (10.4%), and the evaluation is limited to a single model (Mimo-VL-7B), making it unclear whether the benefits generalize.

4. Lack of diversity analysis and potential biases: While Table 1 claims 1,500+ environments, there is no analysis of the actual distribution of applications, platforms, or task types. The dataset is likely heavily skewed toward popular English-language applications from tutorial videos, raising serious concerns about representativeness and potential biases that could propagate to downstream agents.

---

> ### Author Rebuttal · Authors · 2026-03-30
>
> We thank Reviewer XfPh for the detailed feedback. We address the key concerns below.
>
> ---
>
> > ## Q1: Reproducibility and methodological novelty
>
> Thank you for raising these concerns. Our main contribution is Video2GUI, a scalable, top-down pipeline for extracting GUI trajectories from raw YouTube videos. Unlike VideoAgentTrek, which relies on seed keywords and CUA trajectory extraction (hard to scale and prone to bias), Video2GUI enables large-scale dataset construction across diverse platforms through direct filtering and annotation. The Video2GUI pipeline supports both closed- and open-source models. While Gemini-3-Pro achieves higher annotation quality, we also evaluate open-source Qwen3.5-397B-A17B (~15–20% lower, mainly in temporal alignment and spatial grounding). Both the WildGUI dataset and the Video2GUI framework will be open-sourced to benefit the research community.
>
> > ## Q2: Data quality validation
>
> We extend evaluation to 600 manually checked samples across three platforms (Website, Mobile, Desktop) with 200 each, and analyze grounding accuracy across five action types (120 samples each):
>
> | Action Type | Samples | Accuracy |
> |-------------|---------|----------|
> | Click       | 120     | 96.7%    |
> | Scroll      | 120     | 93.3%    |
> | Input/Write | 120     | 91.7%    |
> | Drag/DragTo | 120     | 94.2%    |
> | Press       | 120     | 93.3%    |
> | Overall | 600 | 93.8% |
>
> Furthermore, the significant improvements on ScreenSpot-Pro and OSWorld-G (Table 2) after continual pre-training on WildGUI also confirm reliable localization quality, as these benchmarks directly evaluate precise UI element grounding. We will also add failure case analysis in the revision to characterize common error modes.
>
> > ## Q3: Comparison with VideoAgentTrek and SeeClick
>
> For trajectory quality, human evaluation (Figure 4) confirms WildGUI significantly outperforms VideoAgentTrek in accuracy and task relevance. For downstream performance, we train VideoAgentTrek-CUA-7B with the same post-training data, achieving 37.3 on ScreenSpot-Pro and 58.9 on CAGUI, weaker than WildGUI-pretrained models. CAGUI results demonstrate that WildGUI enhances capabilities in mobile and Chinese scenarios. We also test additional baselines: SeeClick scores 1.1 on ScreenSpot-Pro, Step SR 59.1/75.0 for AndroidControl-High/Low; for proprietary models, Claude achieves 65.3 on AndroidControl-High and 53.8 on CAGUI, all weaker than Mimo-VL-7B + WildGUI. These results demonstrate WildGUI's superiority over VideoAgentTrek in downstream performance.
>
> > ## Q4: Diversity analysis and potential biases
>
> We have provided detailed platform distribution in Figure 6 and application/website category breakdowns. WildGUI covers multiple platforms and includes tutorials in languages such as Chinese and Japanese. Gains on the Chinese-language CAGUI benchmark (+7.6 Step SR) demonstrate that WildGUI can also improve model performance in non-English scenarios. We will add this analysis to the revision.
>
> > ## Q5: Distribution shift between YouTube tutorial videos and real-world GUI usage scenarios
>
> Our pretraining gains transfer to real-world tasks, confirmed from multiple perspectives. First, thanks to the scale of our filtering pipeline (500M metadata entries), WildGUI includes real software usage workflows and complex task demonstrations, not just step-by-step tutorials. Second, consistent improvements on both offline (AndroidControl, CAGUI) and online (AndroidWorld, OSWorld) benchmarks, which involve open-ended, dynamically constructed tasks, provide empirical evidence of transferability.
>
> > ## Q6: API cost for Video2GUI pipeline
>
> The Video2GUI pipeline supports both closed-source and open-source models. Gemini-3-Pro achieves higher annotation quality: for trajectory extraction, each sample requires ~15,908 input tokens, 1,338 output tokens, and 1,452 thinking tokens, costing ~\\$0.0653 per sample. Action grounding adds around \\$0.011 per sample, bringing the total cost to ~\\$0.0763 per sample. For reference, the open-source Qwen3.5-397B-A17B is free to use but results in a 15–20% drop in quality, primarily in temporal alignment and spatial grounding. It is worth noting that dataset construction is a one-time cost. We will release the high-quality dataset along with the reproducible Video2GUI framework to the research community.
>
> > ## Q7: Sensitivity to filtering model choice
>
> As in Appendix A, we fine-tune Qwen2.5-7B as the metadata filter using DeepSeek-V3 annotated labels. Importantly, after coarse filtering, we apply fine-grained video quality scoring (Appendix B). The Qwen2.5-Omni scorer shows high consistency with Gemini-3-Pro annotations (Figure 5), and manual checks confirms strong correlation between scores and actual quality. Thus, the coarse filtering model has limited impact on final data quality. For the threshold, we examine different values and select ≥4.2 on all three dimensions to balance quality and quantity (details in Appendix B).

---

> > ### Author Rebuttal · Reviewer_XfPh · 2026-04-03
> >
> > The authors have addressed most of my concerns, except W1, though I acknowledge this is somewhat beyond the paper's scope as a dataset contribution. I will maintain my positive evaluation and look forward to the final version.

---

### Official Review · Reviewer_Liyk · 2026-03-13

**Soundness:** 3
**Presentation:** 4
**Significance:** 4
**Originality:** 3
**Overall Recommendation:** 5
**Confidence:** 3

**Summary:**

In this paper, the authors propose Video2GUI, an automated and scalable framework that extracts GUI interaction trajectories directly from unlabelled internet videos. Using this Video2GUI pipeline, the authors filter 500 million internet video entries to construct WildGUI, a massive-scale GUI dataset with 12 million trajectories spanning 1,500 applications and websites. The authors leverage WildGUI for pre-training on Qwen2.5-VL and Mimo-VL with additional post-training before evaluating on multiple offline and online agent evaluation benchmarks for both mobile and desktop applications. Experimental results demonstrate that pre-training on WildGUI offers 5-20% improvements across multiple GUI grounding and navigation benchmarks.

**Compliance With Llm Reviewing Policy:**

Affirmed.

**Final Justification:**

My final recommendation is an Accept (Score 5). The WildGUI dataset and the Video2GUI collection method will be helpful resources to the research community. The authors have also sufficiently addressed my concerns regarding human evaluation agreement measurement and the lack of analysis for the GUI agent performance drop when Stage 2 training is removed.

**Key Questions For Authors:**

1. Why was Gemini 3 Pro (a proprietary model) used for trajectory extraction and action grounding, and would it be possible to get an approximate API cost for running the full Video2GUI pipeline? I find this important because it would be useful to know if the same data quality can be achieved with a comparable open-source model and whether the total cost is enough to not be prohibitively expensive.
2. In the human evaluation (Section 5.3), what constitutes a data point, is it a single action step or full trajectory? How were the five expert participants recruited, and what qualifications defined them as experts? The current description makes it somewhat difficult to assess the reliability of the quality claims. I would find the human evaluation much stronger if authors could provide any clear rubrics, evaluation criteria, and perhaps even agreement metrics.

**Limitations:**

No. The Impact Statement could be improved by acknowledging the potential good impact and misuses of GUI agents.

**Strengths And Weaknesses:**

# Strengths:
## Soundness
* The authors validate the usefulness of training with the WildGUI dataset through extensive experiments on GUI grounding, offline and online GUI agent evaluation.
* These evaluations show the two-stage training contributes to large performance improvements of base models Qwen-2.5-VL-7B and Mimo-VL-7B on GUI grounding (ScreenSpot, OSWorld-G), offline evaluation (CA-GUI, AndroidControl), and online evaluation (AndroidWorld, and OSWorld).
* The scaling analysis shows that task accuracy has a strong correlation with increasing the scale of pretraining tokens. The graphs in Figure 3 provide a useful visual for when pre-training data contributes to GUI grounding performance.
* The ablation study is well constructed and shows both how each training loss and stage affects performance on grounding, offline, and online evaluation.
## Presentation
- The paper provides clear technical detail on how the different modules of the Video2GUI pipeline function.
- The two-stage training is clearly explained, and results are neatly displayed in tables for easy lookup.
- This paper addresses the problem of the lack of high quality, diverse, and large-scale GUI trajectory data for training GUI agents.
## Significance
- The Video2GUI pipeline enables the ability to leverage existing web videos to extract good quality GUI data.
- The evaluations and scaling analysis demonstrate that pre-training with this type of data is helpful for training GUI agents.
- The use of existing web demonstration videos for data collection could potentially be applied to other ML domains, and I foresee the release of this pipeline and large scale dataset will be valuable for the research community.
## Originality
- Video2GUI extends existing GUI data collection approaches that leverage web videos (Jang et al. 2025; Song et al. 2025).
- The large-scale video metadata filtering pipeline is a novel introduction and is an improvement over the keyword-based retrieval used by prior works.
- Furthermore, this method enables massive-scale collection of GUI trajectories from many diverse platforms.

# Weaknesses:
## Soundness
* The modest gains (AndroidControl-Low) and decrease (AndroidControl-High) of Qwen2.5-VL-7B + WildGUI for "Type Acc." should be acknowledged and analyzed more thoroughly.
* While the paper acknowledges the large performance drop when "Stage 2" training is removed, I believe it would be strengthened by more analysis regarding this dramatic decrease (especially for the 6% SR on AndroidWorld).
* For human evaluation (Section 5.3), the number of samples reviewed is quite low, 100 manually checked samples compared to the millions of trajectories.
* Some more detail should be given describing the participants and how they could be classified as experts.
* It would be beneficial if we could know how accuracy, diversity, and relevance criteria were defined for participants.
* It would also be helpful to know the cost to produce WildGUI from the pipeline, especially given the use of Gemini-3-Pro for data filtering and action labelling.
## Presentation
- I see a potential discrepancy in one of the AndroidWorld SR number between Figure 2 and Table 4. The AndroidWorld SR for "Stage 2 Only" in Figure 2 shows 23.3 SR%, while the "w/o Stage 1" row for AndroidWorld shows 22.4% SR.
- While the performance increase from training on WildGUI is promising the absolute performance levels for online agent results are still somewhat low (12.3% OSWorld and 31.9% AndroidWorld SR). This shows there is still a large margin for improvement.
- One nitpick: in Appendix F, Rico is mentioned but the citation does not include the original paper (Deka et al., 2017).
## Significance
- While the performance increase from training on WildGUI is promising the absolute performance levels for online agent results are still somewhat low (12.3% OSWorld and 31.9% AndroidWorld SR). This shows there is still a large margin for improvement.

# References
- Jang, Y., Song, Y., Sohn, S., Logeswaran, L., Luo, T., Kim, D.-K., Bae, K., and Lee, H. Scalable video-to-dataset generation for cross-platform mobile agents. In Proceedings of the Computer Vision and Pattern Recognition Conference, pp. 8604–8614, 2025.
- Song, C. H., Song, Y., Goyal, P., Su, Y., Riva, O., Palangi, H., and Pfister, T. Watch and learn: Learning to use computers from online videos. arXiv preprint arXiv:2510.04673, 2025.

---

> ### Author Rebuttal · Authors · 2026-03-30
>
> We thank Reviewer Liyk for the thorough and constructive review. We address each concern below.
>
> ---
>
> > ## Q1: Why Gemini-3-Pro and API cost?
>
> To assess the practical reproducibility of our approach, we estimate the API cost for running the full Video2GUI pipeline. For each sample, trajectory extraction averages approximately \\$0.0653, and action grounding costs approximately \\$0.011, bringing the total to roughly \\$0.0763 per sample. Video quality scoring relies on self-deployed open-source models at negligible cost.
>
> We choose Gemini-2.5-Pro for its superior long-context video understanding and spatial grounding. We also evaluate Qwen3-235B-A22B as an open-source alternative; while functional, it shows ~15–20% lower annotation quality in temporal alignment and spatial grounding. To ensure high data quality at scale, we adopt Gemini-2.5-Pro as the default backbone. A detailed discussion of model selection will be provided in the appendix. Importantly, this is a one-time cost to produce the dataset, and we will release WildGUI publicly, eliminating the need for others to re-run the pipeline.
>
> > ## Q2: Human evaluation details
>
> Each data point corresponds to a single trajectory, not an individual action step. We recruit several CS masters/Ph.D. candidates with prior VLM-based GUI agent research experience and not involved in this project. They receive detailed guidelines on GUI trajectory formats and quality standards. Each completes a 20-sample trial; only those achieving ≥0.85 accuracy qualify, yielding five expert participants.
>
> Evaluation criteria: (1) Accuracy measures whether actions are correctly identified with proper timestamps and spatial coordinates; (2) Diversity measures richness of platforms and task types within randomly sampled data; (3) Relevance evaluates whether trajectories reflect meaningful, real-world GUI tasks.
>
> We compute Krippendorff's α across the five evaluators, obtaining α = 0.84, indicating good agreement. We will include detailed rubrics and the agreement analysis in the revision. We also acknowledge that 100 samples is limited and will expand to 300 samples across platforms. Furthermore, the performance gains after training on WildGUI, as well as improved scores on ScreenSpot-Pro compared to VideoAgentTrek-CUA-7B's 37.3 (trained on VideoAgentTrek data), provide additional evidence of our dataset's quality.
>
> > ## Q3: AndroidControl Type Acc. for Qwen2.5-VL
>
> Thanks for the comment. We agree that the modest gain on AndroidControl-Low Type Acc. and decrease on AndroidControl-High Type Acc. warrant further analysis. As we analyze, the limited improvement on the low-level setting is likely due to the model already achieving a high baseline (94.1%), leaving little room for further gains. The minor drop on the high-level setting may stem from the distribution shift introduced by WildGUI, which contains diverse action types across platforms. Since action type prediction is a simple classification task that saturates quickly, these fluctuations are expected. Importantly, the substantial improvements in Step SR, which demands both accurate action type and precise spatial grounding, demonstrate the overall effectiveness of our approach. We will incorporate this analysis into the paper.
>
> > ## Q4: Stage 2 performance drop analysis
>
> We will follow your suggestion to strengthen the analysis and clarify the necessity of Stage 2 training. The drop without Stage 2 (e.g., 6.0% on AndroidWorld) stems from a mismatch: WildGUI pre-training provides diverse GUI data, but its format differs from online environments in action spaces and history structures. Inspecting model trajectories reveals that without Stage 2, the model often generates unsupported actions (e.g., desktop hotkeys in mobile settings or incompatible parameter formats), leading to cascading failures from which the agent cannot recover. Stage 2 post-training can effectively align the model's output with the target environment's action space and interaction protocol.
>
> > ## Q5: Absolute performance levels for online agent results are somewhat low
>
> We agree that the absolute scores of OSWorld and AndroidWorld leave room for improvement. Online environments pose unique challenges including dynamic state transitions, error recovery, and long-horizon planning beyond what pre-training alone can address. Higher performance likely requires environment-specific post-training data and online RL. In future work, we plan to explore online RL fine-tuning and environment-specific data collection. Notably, WildGUI already provides a strong foundation, nearly doubling AndroidWorld SR from 16.4% to 31.9%.
>
> > ## Q6: Presentation and Reference
>
> Thanks for the correction. We will fix the Figure 2 vs Table 4 discrepancy, add the Rico citation [1], expand the Impact Statement to discuss both positive impacts and potential misuses of GUI agents.
>
> ---
>
> **References**
>
> [1] Rico: A mobile app dataset for building data-driven design applications

---

> > ### Author Rebuttal · Reviewer_Liyk · 2026-04-03
> >
> > I thank the authors for the detailed rebuttal, which has adequately addressed my concerns, especially regarding human evaluation. I will maintain my positive review and look forward to seeing these revisions added to the final version.

---

### Decision · Program_Chairs · 2026-04-30

**Decision:**

Accept (regular)

**Comment:**

This paper introduces Video2GUI, an automated framework for extracting GUI interaction trajectories from unlabeled web videos, and presents WildGUI, a massive-scale dataset comprising over 12 million trajectories. The reviewers unanimously agreed that the paper tackles a critical data scarcity bottleneck in training generalized GUI agents, praising the dataset's unprecedented scale, diversity, and the solid 5–20% performance improvements it yields on downstream grounding and navigation benchmarks. During the review phase, reviewers raised concerns regarding the initially small sample size for human evaluation of data quality, the reliance on proprietary models (Gemini-3-Pro) for trajectory extraction, and the need for comparisons against stronger or more recent baselines. In a highly effective rebuttal, the authors successfully addressed these issues by expanding their manual quality evaluation to 600 samples with detailed action-type breakdowns, transparently documenting API costs while exploring open-source model alternatives, and providing new empirical results demonstrating that WildGUI pre-training enhances even stronger, state-of-the-art baselines (e.g., Qwen3-VL-8B) while outperforming contemporary datasets like VideoAgentTrek. Given the rigorous validation provided in the rebuttal, the unanimous satisfaction of the reviewers, and the significant practical value this open-sourced pipeline and dataset will provide to the GUI agent research community, the paper is recommended for acceptance.